# Exponential Dynamic Energy Network for High Capacity Sequence Memory

**Arjun Karuvally**
Salk Institute for Biological Studies
akaruvally@salk.edu

**Pichsinee Lertsaroj**
University of Massachusetts Amherst

**Terrence J. Sejnowski**
Salk Institute for Biological Studies
terry@salk.edu

**Hava T. Siegelmann**
University of Massachusetts Amherst
hava@umass.edu

## Abstract

The energy paradigm, exemplified by Hopfield networks, offers a principled framework for memory in neural systems by interpreting dynamics as descent on an energy surface. While powerful for static associative memories, it falls short in modeling sequential memory, where transitions between memories are essential. We introduce the Exponential Dynamic Energy Network (EDEN), a novel architecture that extends the energy paradigm to temporal domains by evolving the energy function over multiple timescales. EDEN combines a static high-capacity energy network with a slow, asymmetrically interacting modulatory population, enabling robust and controlled memory transitions. We formally derive short-timescale energy functions that govern local dynamics and use them to analytically compute memory escape times, revealing a phase transition between static and dynamic regimes. The analysis of capacity, defined as the number of memories that can be stored with minimal error rate as a function of the dimensions of the state space (number of feature neurons), for EDEN shows that it achieves exponential sequence memory capacity $\mathcal{O}(\gamma^N)$, outperforming the linear capacity $\mathcal{O}(N)$ of conventional models. Furthermore, EDEN's dynamics resemble the activity of time and ramping cells observed in the human brain during episodic memory tasks, grounding its biological relevance. By unifying static and sequential memory within a dynamic energy framework, EDEN offers a scalable and interpretable model for high-capacity temporal memory in both artificial and biological systems.

## 1 Introduction

Memory is a crucial element of cognition that is essential for learning, reasoning, and decision-making. Understanding and replicating the human ability to store and recall information is a long-term challenge in both biological and artificial intelligence (AI). The energy paradigm, introduced by Hopfield and Amari 40 years ago, revolutionized memory modeling by characterizing the dynamical behavior of neural networks using an energy landscape [1, 2]. According to the energy paradigm, a stimulus instantiates a network state on an energy landscape. The neurons then interact with each other such that the state travels down the landscape until a minimum is reached. This minimum state is defined as the memory of the network. The energy approach to memory modeling represented a significant advancement of our scientific understanding of memory by offering an intuitive understanding of network dynamics, with added theoretical guarantees of stability. The disadvantage was that the number of memories that can be reliably stored was only a small fraction of the number of neurons [3, 4, 5] and scaled linearly with the increase in neurons. This limited the applicability of energy

39th Conference on Neural Information Processing Systems (NeurIPS 2025).

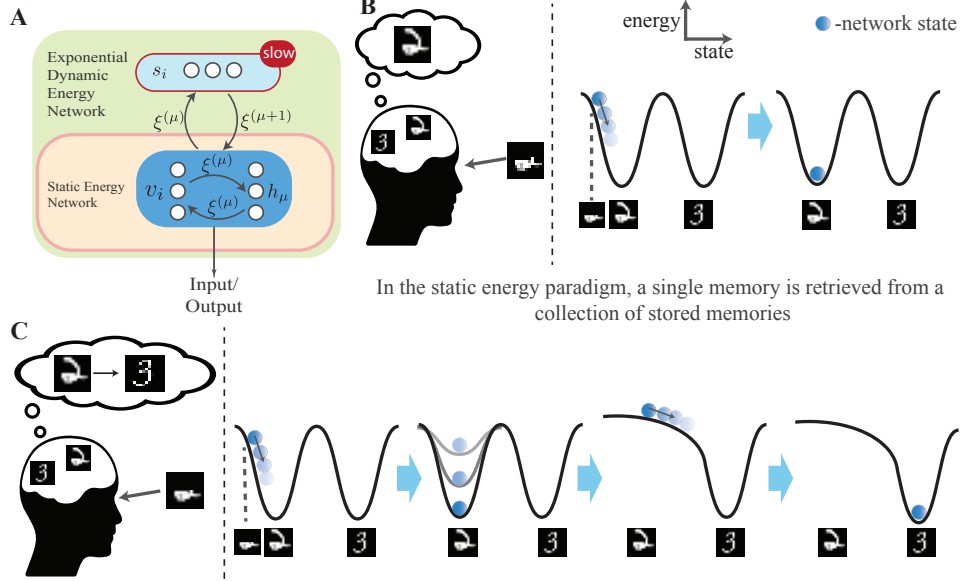

Figure 1: **Schematic Model and Energy Landscape Behavior of Dynamic Energy Networks**:
**A.** The dynamic energy network, EDEN, has asymmetrically interacting slow neurons providing information about the next memory in the sequence to the two fast neural populations. **B.** Static energy-based networks are used as models of human associative memory where a single memory associated with a provided stimulus is recalled. EDEN, without the slow population, is a static energy network that retrieves a single memory from a collection of stored memories. The system's state $(v_i)$, represented by the blue ball, descends the energy surface until a stable memory (energy minimum at state "2") is reached. After retrieval, the state of the system does not change and stays at "2". **C.** Dynamic energy networks enable associative *sequence* memory, where the associated memory along with its sequential neighbors are recalled. In EDEN, the energy surface changes in response to the state of the system, causing the minima of the energy surface to change over time (from "2" to "3"), resulting in transitions between memories.

networks despite their theoretical advantages. Further, the dynamics on the energy surface guaranteed a *single* final memory, precluding any temporal behavior in the memories. Further research in improving these networks proceeded in two independent directions. In one direction, researchers sought to develop techniques to improve the limited memory capacity of the original neural network by proposing modifications to how neurons interact in the network. In the second direction, researchers sought to create alternative formulations to energy function such that sequences and temporal aspects of memory can also be modeled with similar theoretical guarantees.

Improving capacity has been central to the development of memory models. In the context of Hopfield networks, capacity is defined as the maximum number of memories that can be stored with minimal errors as a function of the number of dimensions in its state space (the number of visible neurons). Earlier studies revealed that the limited capacity of the classic Hopfield network was due to significant crosstalk between the memories resulting in energy functions with many spurious minima. A major breakthrough in capacity came with the introduction of higher order terms in the energy function that separated the contribution of each memory to the energy minimum [6, 7, 8, 9, 10, 11, 12] resulting in polynomial capacity scaling and dense networks - networks that store more memories than the number of neurons [13]. Further studies introduced exponential terms, greatly increasing memory capacity and enabling practical applications [14, 13]. Currently, energy networks are used in AI with applications in large-scale natural language processing [15, 16], computer vision [17], and lifelong-learning systems [18, 19] as reliable external memory storage. Further, the self-attention mechanism in transformer architectures have been shown to be functionally equivalent to the exponential memory capacity network providing insights into their mysterious capabilities [20, 21]. The success story of high-capacity static energy networks demonstrates how utilizing the energy paradigm benefits advancement and practical applications.

Despite these advancements in the energy paradigm, state-of-the-art networks are still restricted to retrieving single memories from a collection of stored memories. Reconciling the single stable memory states in the energy paradigm with the dynamic states required for modeling sequences remains a significant challenge [22, 23, 24, 25]. Over the years, there have been several solutions proposed for the challenge. One proposal introduced networks combining symmetric interactions, asymmetric interactions, and delay signals to produce temporal behavior [26, 27]. These proposals succeeded in creating networks that exhibited sequential state transitions, but the energy paradigm could not be applied to these cases as occasionally the network traveled up the energy surface. Another proposal introduced noise into the dynamics for enabling transitions out of a memory basin of the energy surface [28, 29, 30, 31, 32, 33]. The energy paradigm applied to these models revealed a lowering of the energy barrier between states as more noise is added to the system. Without theoretical insights obtained from the application of the energy paradigm, the modifications needed to improve sequence capacity could not be found. As a result, extant sequence networks have capacity much lower than the number of neurons. Developing an energy principle for temporal memory networks will enable memory researchers to develop networks that are capable and aligned with experimental data. It will also enable artificial intelligence researchers to develop capable external memory stores.

Our work extends the energy paradigm to temporal memories by allowing the energy surface to change slowly with time. This approach was previously proposed experimentally in [34] and some computational properties studied in [35]. In contrast to the classical energy paradigm, the memories in the dynamic energy networks can lose or gain stability over time, resulting in stability in two timescales. In short timescales, the current memory is always stable, with the energy function guaranteeing convergence and robustness to noise. In longer timescales, the energy surface changes to create a new minimum, destroying the current minimum. The network state changes in response, resulting in stable transitions between memory states. Our analysis of the proposed dynamic energy network shows that (1) The network's dynamical behavior is well characterized by the short-timescale energy functions assembled piecemeal for long-timescale dynamical behavior, (2) The energy function provides a precise analytic computation for the time required to escape from a stable memory state and the conditions necessary to exhibit memory transitions, (3) The network capacity scales exponentially in the number of neurons, significantly outperforming existing sequence memory networks, (4) The network populations have biological implications, showing strong behavioral correlations to the activity of cells found in human episodic memory experiments. The new paradigm thus enables the development of biologically relevant sequence memory networks with improved storage capacity.

Our work also provides theoretical insights into current approaches to sequence memory modeling. Notably, we extend our earlier work on sequence memory [34] with theoretical analysis about dynamical behavior, and rigorous claims of dense capacity. Another approach used in [36] has similar multiple-timescale dynamics where the sequences are learned from the stimulus and the transitions are governed by successive bifurcations. A more recent work [37] introduced a similar softmax function with asymmetric synapses for dense capacity in a discrete network without using the energy arguments. Our work reveals that the successive bifurcations hypothesized by [36] are due to the change in stability of the energy landscape, and the capacity increase observed by [37] may be due to separating the memory contributions to the energy functions.

## 2 Results

### 2.1 Exponential Dynamic Energy Network (EDEN)

To develop dynamic energy networks with exponential capacity, we incorporated a slow-changing signal that interacts asymmetrically with an exponential capacity static energy network introduced in prior research [21]. The resulting model is a system of interacting neurons with slow and fast timescale neural populations. The slow timescale population modulates the energy surface for the fast timescale population resulting in a system with a temporally varying energy function.

Mathematically, our model is a two-population neural network. The first population consists of a feature layer (input/output layer) represented by the vector $v$ and a hidden layer represented by the vector $h$. There are $N$ neurons in the feature layer and $P$ neurons in the hidden layer (one for each memory that needs to be stored in the network). This two-layer organization of the fast networks is primarily motivated by a recent general theory of energy-based networks [13]. The feature and hidden layer make the fast timescale population and are part of the exponential capacity static energy network.

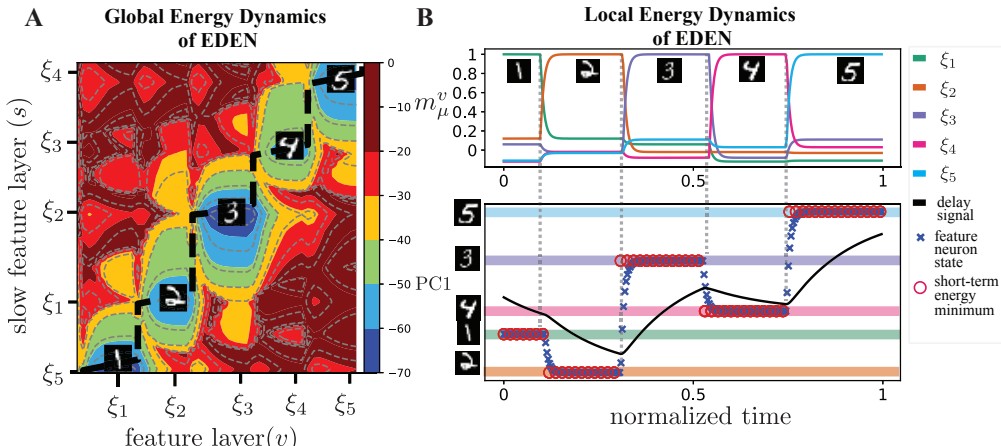

Figure 2: **Simulation of EDEN reveals robust transitions between memory states and the existence of local energy functions**: EDEN is simulated to store and retrieve a simple sequence of 5 MNIST digits in numeric order. **A** The global energy surface with both slow and fast populations of EDEN shows the neural state traversing a valley of the energy surface with occasional energy-increasing regimes. **B.** The dynamical behavior of the memory overlaps of the fast population ($m_\mu^v = \frac{1}{N} \sum_{i=1}^{N} v_i \xi_i^{(\mu)}$) of EDEN and the analysis of the first principal component (PC1) of the time evolution of its fixed points show the fast population (blue cross) converging to the instantaneous minimum of short-timescale energy functions (red circles). The short-timescale energy minimums are modulated by the slow population. As the slow population approaches the current state of the fast population, the energy minimum switches to the sequentially connected memory. Over time, these short-timescale energy changes slowly so that the fast population has sufficient time to relax at its instantaneous minimum. The long-timescale dynamical behavior of the network can then be assembled from the short-timescale behaviors.

In the fast population, the hidden layers are instantaneous (very fast) enabling rapid information transfer and follow the state of the art practices in developing energy networks. The interaction between the feature neuron $i$ and the hidden neuron $\mu$ is symmetric and is represented by the synaptic weight $\xi_{i\mu}$. The vector obtained by $\xi_i^{(\mu)}$ for a fixed $\mu$ and $i \in \{1, 2, \ldots N\}$ is the $\mu^{\text{th}}$ stored memory (energy minimum) of the system. We analyze the network in the paper under the assumption of Rademacher distributed memory patterns - $\Pr\left[\xi_i^{(\mu)} = +1\right] = \Pr\left[\xi_i^{(\mu)} = -1\right] = 1/2$.

The population of slow neurons represented by the vector $s$ is the continuous delay signal from the feature neurons. Therefore, there are $N$ delay neurons. This slow signal retains information about the *previous* memory state with a characteristic dynamical timescale - $\mathcal{T}_d$. We consider the case when the timescale of the slow neurons is higher compared to the feature neurons ($\mathcal{T}_d \gg \mathcal{T}_f$). This timescale difference enables the existence of short timescale energy functions. The neurons in the slow population interact with the hidden layer neurons through the synapses represented by the vector $\xi^{(\mu-1)}$. For simplicity, we assume the memories are arranged in a single long circular sequence with $\xi^{(\mu-1)} \to \xi^{(\mu)}$ for $\mu > 1$ and $\xi^{(P)} \to \xi^{(1)}$, where $P$ is the number of memories in the sequence to be stored. For exponential memory capacity scaling, the softmax activation function is used for the hidden layer. The evolution of the resultant network is given by the following set of mathematical equations with Latin characters indexing the feature and slow neurons, and the Greek characters

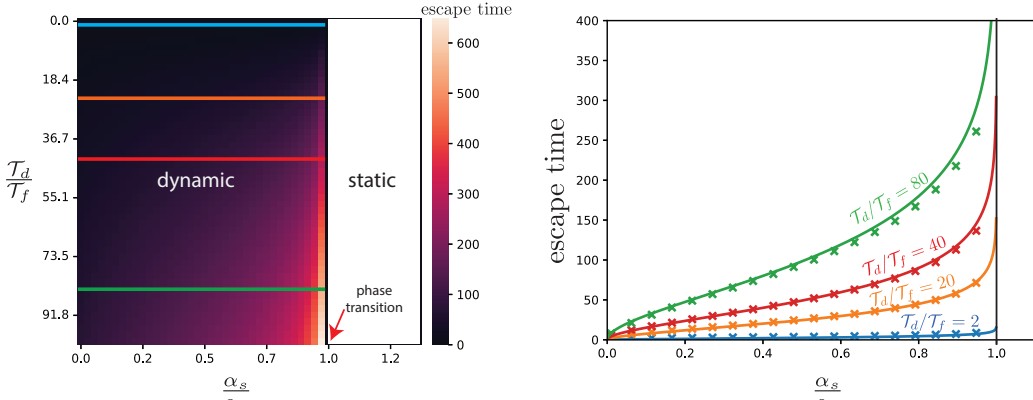

**Escape Time Characteristics of EDEN**

Figure 3: **Escape Time Characteristics of EDEN under different parameter regimes**: (**left**) The analysis of the escape times (in $\mathcal{T}_f$ units) of EDEN under different parameter settings shows two different dynamic regimes. When the coefficient ratio $\alpha_s/\alpha_c > 1$, EDEN has static memories where the dynamic behavior converges to one of the stored memories without any transitions. When the coefficient ratio $\alpha_s/\alpha_c < 1$, EDEN has memory transitions. (**right**) We take 4 sample cross sections of the phase diagram, shown by the colored horizontal lines on the left. The average time required to escape a memory state is characterized by the timescale ($\mathcal{T}_d/\mathcal{T}_f$) and the coefficient ($\alpha_s/\alpha_c$) ratios. The analytical escape times (the solid lines) computed from the energy function show good agreement with the experimental values (the points) with a mean absolute error of $5.96\mathcal{T}_f$ units.

indexing the hidden layer neurons.

$$
\begin{cases}
\mathcal{T}_f \dfrac{\mathrm{d}v_i}{\mathrm{d}t} &= \displaystyle\sum_{\mu=1}^{P} \xi_i^{(\mu)} \frac{\exp(h_\mu)}{\sum_\nu \exp(h_\nu)} - v_i \,, \\[2ex]
h_\mu &= \alpha_s \displaystyle\sum_{i=1}^{N} \xi_i^{(\mu)} v_i + \alpha_c \sum_{i=1}^{N} \xi_i^{(\mu-1)} s_i \,, \\[2ex]
\mathcal{T}_d \dfrac{\mathrm{d}s_i}{\mathrm{d}t} &= v_i - s_i \,.
\end{cases}
\tag{1}
$$

The interaction strength coefficient for the self-memory interaction is $\alpha_s$ and for cross-memory interaction is $\alpha_c$. The self-memory interactions connects a memory with itself ($\xi^{(\mu)}$ with $\xi^{(\mu)}$), stabilizing the current memory of the network. The cross interactions drive the asymmetric interactions ($\xi^{(\mu-1)}$ with $\xi^{(\mu)}$) which causes state transitions. This dynamical system of interacting neurons has the following energy function for the fast population (Appendix B).

$$
E(v) = \underbrace{\sum_{i=1}^{N} \frac{(v_i)^2}{2}}_{\text{state energy}} - \underbrace{\frac{1}{\alpha_s} \cdot \log\left(\sum_{\mu=1}^{P} \exp\left(\alpha_s \sum_{i=1}^{N} \xi_i^{(\mu)} v_i + \alpha_c \sum_{i=1}^{N} \xi_i^{(\mu-1)} s_i\right)\right)}_{\text{interaction energy}} .
\tag{2}
$$

The first term represents the state energy of the network, and the second term represents the interaction energy from the synapses. The interaction energy now contains additional terms for the slow population compared to the energy function of a typical Hopfield-type network. The interaction energy from the fast population generates minima near a *similar* memory (defined as the memory with the most overlap $m_\mu^v$), while the slow population generates minima near the sequentially connected memory. The dynamical behavior of the overall system is characterized by the relative strengths of these two interaction terms. With the network's dynamics defined, we now analyze how its behavior differs from that of static energy networks.

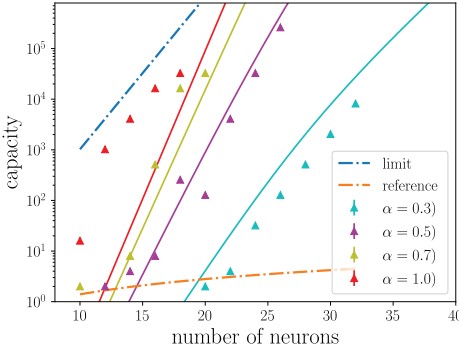

Figure 4: **Exponential Sequence Memory Capacity of EDEN**: The plot shows the fixed point capacity in the $\log_{10}$ scale for EDEN simulated with different $\alpha_c = \alpha$ (with $\alpha_s = 0.999\,\alpha$) compared with the reference network when small errors ($\delta < 10^{-3}$) are tolerated. The analytic curves are shown as solid lines and experimental values as points. The reference network capacity scales linearly with the asymptotic rate of $O(N)$ (dotted orange line), while EDEN scales *exponentially* with the asymptotic rate $O(\gamma^N)$ in the number of feature neurons. The exponent base is higher than the limit ($\gamma > 2$), enabling EDEN to reach the available capacity limits of $2^N$ (dotted blue line) in the asymptotic limit of the number of neurons.

## 2.2 Energy Dynamics

Conventionally, an energy function precludes any temporal memory behaviors, as the dynamic requirements of temporal memories conflict with the convergent dynamics found in systems with an energy function. However, this argument assumes that the energy function that characterizes the behavior of a system is constant. The theoretical analysis of EDEN reveals that the long-term dynamical behavior of the network can be well explained piecemeal by short-term energy functions.

To analyze how the dynamics of the energy change with the introduction of the slow population, we take the time derivative of the energy function from Equation 2 along the dynamical trajectory of the system. The dynamical evolution of the energy function after separating the two timescales is shown below (see Appendix B for the full derivation).

$$
\frac{\mathrm{d}E}{\mathrm{d}t} = -\mathcal{T}_f \underbrace{\sum_i \left(\frac{\mathrm{d}v_i}{\mathrm{d}t}\right)^2}_{\text{fast timescale }(F)} - \frac{\alpha_c}{\alpha_s} \underbrace{\sum_{i,\mu} \frac{\exp(h_\mu)}{\sum_\kappa \exp(h_\kappa)} \xi_i^{(\mu-1)} \frac{\mathrm{d}s_i}{\mathrm{d}t}}_{\text{slow timescale }(S)} \ . \tag{3}
$$

The two terms of Equation 3, which we label by $F$ and $S$ separate the contributions of the fast and slow timescales. Excluding the $S$ term, the fast population will have one of two possible behaviors. When the sign of $F$ is negative, the population converges to a single stable state corresponding to the minimum of the energy function. When the term is $0$, the system moves in an iso-energetic (states that have the same energy) trajectory without convergence. In this paper, we focus only on the case of convergent behavior. We find that the case of non-convergence does not arise in the simulations.

The slow population influences the second term, $S$. When the slow timescale is longer compared to the fast timescale (under the condition that $\mathcal{T}_d \gg \mathcal{T}_f$), which we assume in the paper, the network exhibits a non-increasing energy function and the effect of $S$ is effectively negligible. The analysis reveals two roles the slow population plays in the network dynamics: (1) The slow dynamical nature helps to *stabilize* the dynamics of the fast population on the energy surface, enabling it to converge to a memory state (2) The asymmetric interactions of the slow population *changes* the energy surface to create new minima and destroy old minima, inducing memory transitions. These two functions result in a network with stable transitions between memories.

In our numerical simulations, we consider settings of the slow timescale to be high enough for the slow neurons to change sufficiently slowly for the energy function to characterize the dynamics but not so high as to prevent the system from exhibiting state transitions in a reasonable time. Figure 2

shows the energy function behavior of EDEN and the dynamic behavior of the feature to memory overlaps $m_\mu^v = \sum_i \xi_i^{(\mu)} v_i$. The analysis reveals that although a single energy function does not characterize the global temporal behavior of the network, the local behavior is well described by short-timescale energy functions. Analysis of the fixed points of the energy surface predicts when an instability leads to memory transition and governs where each memory transitions to next. A global behavioral characterization can then be obtained piecemeal from these local characterizations.

## 2.3 Escape Time Characterization of EDEN

To determine the variety of dynamical behaviors exhibited by EDEN, we analyzed how its parameters - the timescales $(\mathcal{T}_d, \mathcal{T}_f)$ and the interaction strength coefficients $(\alpha_c, \alpha_s)$, influence the escape time - the time the network spends in a memory state before transitioning to the next. To formalize the average escape time, we define that the network state at some time $v(t)$ is in a memory state $\mu$ if $\mu = \arg\max_\nu \sum_i \xi_i^{(\mu)} v_i(t)$, that is, if the $\mu^{\text{th}}$ memory has the maximum overlap with the network state compared to all other memories. Formally,

$$t_e(\mu) = \max\left\{t : \mu = \arg\max_\nu \sum_i v_i(t)\xi_i^{(\nu)}\right\}, \tag{4}$$

when $v(0) = \xi_i^{(\mu-1)}, s(0) = \xi_i^{(\mu-2)}$. The average escape time is defined as the time the network stays in a memory state $\mu$ averaged across all the memories. Computing the escape time for nonlinear dynamical systems like EDEN is a significant challenge. However, since we have access to the system's energy function, we compute escape time analytically using the time required for the energy function to change minima from a memory state $\xi^{(\mu-1)}$ to a memory state $\xi^{(\mu)}$. The escape time is obtained by evaluating the time taken for the energy contribution of $\xi^{(\mu-1)}$ to be lesser than $\xi^{(\mu)}$ when the network initially starts at $\xi^{(\mu-1)}$ and eventually transitions to $\xi^{(\mu)}$. That is, $\exp\left(\alpha_s \sum_i \xi_i^{(\mu-1)} v_i + \alpha_c \sum_i \xi_i^{(\mu-2)} s_i\right) < \exp\left(\alpha_s \sum_i \xi_i^{(\mu)} v_i + \alpha_c \sum_i \xi_i^{(\mu-1)} s_i\right)$. We obtain the following analytic expression for the expected escape time, assuming the effect of transients in the fast population is negligible (details in Appendix D) and that the transitions are Markovian. These assumptions are reasonable, as in the slow timescale limits we consider in the paper, the memory transients are observed to be almost instantaneous relative to the amount of time spent in a memory state (in Figure 2) and the time spent is enough for the network history to decay. The average escape time has the analytic expression given below.

$$\langle t_e \rangle = -\frac{\mathcal{T}_d}{\mathcal{T}_f} \ln\left(1 - \sqrt{\frac{\alpha_s}{\alpha_c}}\right) \tag{5}$$

The phase diagram in Figure 3 constructed from the analytic escape time shows that the ratio of coefficients $\frac{\alpha_s}{\alpha_c}$ uniquely determines the emergence of two different regimes in the dynamical behavior of EDEN. In the static memory regime, when $\frac{\alpha_s}{\alpha_c} > 1$, the cross-interaction strength is weaker than the self-interaction strength, resulting in infinite escape time and EDEN exhibiting the dynamical behaviors of a static energy network. For $\frac{\alpha_s}{\alpha_c} < 1$, the cross-interaction strength is greater, and EDEN enters the dynamic memory regime. The coefficient fraction and the slow-fast timescale ratios define the escape time in the dynamic memory regime. The escape time is sufficiently high for larger timescale ratios to observe stable transitions, making it ideal for storing memory sequences. On the other hand, reducing the slow timescale parameter results in noisy dynamics between memories characterized by short escape times. Ensuring that the coefficients are close to the phase transition boundary enables the resulting network to exhibit stable transitions with a long time spent in memory states.

## 2.4 EDEN has exponential capacity

Now that we have a network that follows an energy function, we evaluate how well the capacity guarantees of the exponential static energy networks translate to the dynamic case. For simplicity, we compute the capacity for networks at the phase change boundary $\frac{\alpha_s}{\alpha_c} \to 1$. The networks at the transition boundary have infinite escape time, resulting in the slow population completely forgetting the previous memory state at the transition point. This enables precisely defining the slow population's state at the transition point. For a network at the phase boundary, the fixed point capacity is defined as

the maximum number of memories that can be stored as a function of the size of the state space ($N$) of the networks. As an added nuance, this ignores the number of hidden neurons in the framework. This follows extant definitions of capacity. Minor errors are allowed in the retrieved memory, with $\epsilon$ defining how close the fixed point is to the target memory state and $\delta$ defining the *rate* of tolerable bit errors. Mathematically, the capacity is defined as

$$C(N, \epsilon, \delta) = \max \left\{ P \in \mathbb{N} : \Pr\left[ v_i(t_e) \cdot \xi_i^{(\mu)} \geq 1 - \epsilon \right] \geq 1 - \delta \right\} \tag{6}$$

with,

$$v(0) = \xi^{(\mu-1)}, \text{ and } s(0) = \sqrt{\frac{\alpha_s}{\alpha_c}} \xi^{(\mu-2)} \tag{7}$$

The factor $\sqrt{\frac{\alpha_s}{\alpha_c}}$ for the slow population was obtained by solving for its state analytically during state transition (Equation 24 in the Appendix). We then compare the fixed point capacity of EDEN with the following reference network.

$$\begin{cases} \mathcal{T}_f \dfrac{\mathrm{d}v_i}{\mathrm{d}t} &= \left( \alpha_s \sum_{\mu j} \xi_i^{(\mu)} \xi_j^{(\mu)} \sigma(v_i) + \alpha_c \sum_{\mu, j} \xi_i^{(\mu)} \xi_j^{(\mu-1)} s_i \right) - v_i \,, \\ \mathcal{T}_d \dfrac{\mathrm{d}s_i}{\mathrm{d}t} &= v_i - s_i \,. \end{cases} \tag{8}$$

where the nonlinearity $\sigma$ is defined as

$$\sigma(x) = \begin{cases} -1 & x < -1 \\ x & -1 \leq x \leq 1 \\ 1 & x > 1 \end{cases} \tag{9}$$

Minor variations of this reference network have been previously studied as multiple timescale models of sequence memory [26, 36, 38], making it suitable as a proxy for existing multiple timescale sequence networks. The notable difference between the reference network and EDEN is the absence of a hidden layer and the softmax activation function. As a result, the reference network has linear interaction between the neurons in the memory layer.

The analytic form for the capacity of the network is obtained from a given $N, \epsilon, \delta$ as (details in Appendix E.2.2)

$$C_{\text{EDEN}} = k(\epsilon, \delta) \left( \frac{\exp(\alpha r) \exp(\alpha)}{\cosh(\alpha r) \cosh(\alpha)} \right)^{N-1}, \tag{10}$$

where $k$ is a constant independent of $N$ in the large $N$ limit. The capacity is exponential in the number of neurons with the asymptotic rate of $O(\gamma^N)$, where $\gamma = \frac{\exp(\alpha r) \exp(\alpha)}{\cosh(\alpha r) \cosh(\alpha)}$. The maximum capacity possible for a network with $N$ neurons is $2^N$, and the exponent $\gamma > 2$ for most choices of $\alpha$ suggests that EDEN reaches the maximum possible capacity in the large $N$ limits. The capacity of the reference network is similarly obtained as

$$C_{\text{ref}}(N, \epsilon, \delta) = N \frac{\epsilon^2 \delta}{\ln(N)} \tag{11}$$

The capacity of the reference network is only linear in the number of neurons. For large $N$, the asymptotic capacity is $O(N)$, which is only linear in the number of neurons. The analytic results are compared against simulations of networks with $N \in \{10, 12...35\}$ in Figure 4. The results show an exponential improvement in the scaling behavior of EDEN when compared to the reference network. Further, EDEN approaches the available limit of $2^N$ memories for higher settings of $\alpha$. Due to computational constraints, the maximum number of memories to be stored was limited to $< 10^6$.

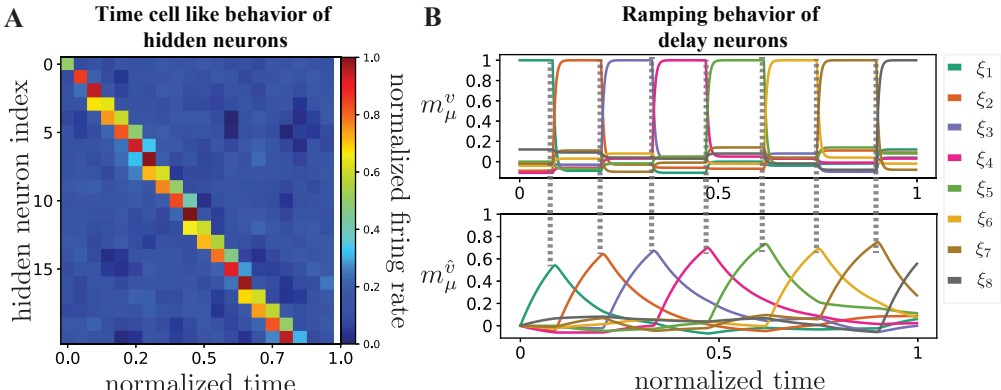

Figure 5: **The EDEN neural populations shows behavioral similarity to cells observed in human episodic memory experiments**: **A** The heatmap of the hidden layer neuron activity ordered by time shows time-sensitive behavior analogous to the time cells observed in human episodic memory retrieval experiments of [39]. **B** The slow layer neurons *ramp up* their activity until it reaches the current memory which in turn induces the transition to the next memory. Rather than an instantaneous drop in their activity, the slow layer slowly *ramps down* to stabilize the feature layer state on the next memory. This ramp up and ramp down activity is analogous to the activity of ramping cells observed in episodic memory experiments [39].

## 3  Biological Relevance

Episodic memory is the human ability to remember when and what happened during specific events through an autobiographical recall of information [40]. Episodic memory is evaluated in humans using list recall tasks [41, 42, 43]. As an essential component of cognition, the role of brain cells in supporting episodic memory is an important question. Experimental studies in human episodic memory have identified time cells and ramping cells in the hippocampus and entorhinal cortex as playing a role in encoding and retrieving episodic memories [39]. Time cells activate in a sequence corresponding to the order of the events being recalled and are hypothesized to encode the temporal information of the recalled memory. Ramping cells also activate to the timing of memories but show only a gradual increase or decrease in activity encoding time in longer timescales. With our theoretical setup for retrieving sequential memories, we can analyze the retrieval aspect of the list recall task. Our findings show that the fast hidden neuron population and the slow population show behavioral characteristics similar to those of the time cells and ramping cells observed in neurobiological experiments supporting episodic memory. This indicates that the EDEN architecture may be used to develop theories and simulate neurons for evaluating episodic memory in the brain.

Figure 5 shows the behavioral correlations between the two populations of neurons in EDEN and the cells found in human episodic memory experiments. Specifically, the dynamic nature of EDEN's hidden neurons lines up sequentially like time cells, reacting to the timing of the stimulus during memory retrieval. The slow neuron population in EDEN shows a gradual rise and fall in activity, analogous to the ramping cells, encoding the timing context during the retrieval of memories. Our theoretical analysis of EDEN shows that the slow population helps stabilize the retrieved memory on the energy surface and directs the transition between retrieved memories. Moreover, the time it takes for memories to shift from one state to another in EDEN is influenced by the ramping cells' timing and the strength of their connections to other neurons. The theoretical insights from EDEN suggest that ramping cells may play a role in stabilizing and directing transitions in addition to simply encoding temporal information as hypothesized from experiments. The time cells, on the other hand, being the fast population only react to the state of the slow population and play a role in identifying and arranging the retrieved memories in time.

## 4  Discussion

The Hopfield-Amari networks and the energy paradigm have provided foundational knowledge of neural networks. However, addressing the diverse behaviors found in neural networks, it is imperative

to evolve the energy paradigm beyond its traditional roots of static memory retrieval. We suggest EDEN as a model that takes a step in this direction by introducing slow-timescale dynamics with asymmetric memory interactions to the energy function, creating a new dynamic energy paradigm. The results point to the enhanced capacity and understanding enabled by the new dynamic energy paradigm. With these results, we posit that the network and theory could shed light on other temporal characteristics of human memory experiments. In addition to the potential impact on neuroscience, the simulations suggest that dense memory may be used in AI applications requiring robust, high-capacity sequence memory storage and retrieval. The proposed energy paradigm provides a universal framework for memory computations in static and dynamic cases. Further, the biological relation of EDEN provides a path for analyzing the episodic memory experiments in a tractable framework that will inform future studies on memory. In future studies, we plan to generalize the energy networks further to complex, realistic sequences and dynamic working memory settings.

**Limitations.** As a theory of dynamical behavior of a non-linear system, we make key assumptions that simplify our mathematical analysis. (1) The synaptic strengths of the neuron interactions are fixed and does not vary during training, in actual biological systems synaptic strengths can change due to short and long term potentiation effects and consolidation (2) The timescales of symmetric and asymmetric interactions are separate - this allows use to treat the asymmetric part as slowly evolving and change the energy function of the symmetric network is response. In human brains, there are different timescales for information processing but the timescales may not be perfectly separated as a distinct slow population of asymmetric connections and fast population of symmetric population, (3) Binary memory - we assume Rademacher distributed patterns for theoretical exposition following related works in the field, although the theory can be similarly worked out for other distributions (4) Markovian State Transition - in deriving our capacity bounds, we assumed that the network spends enough time in a memory state that the historical trajectory information is lost and the state transitions are purely Markovian in nature. Further, the capacity bounds we formulated shows how the maximum number of memories (number of hidden neurons) scales with the number of visual neurons following previous results in the field. The number of hidden neurons required for storage however grows only linearly in the number of hidden neurons.

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

## A  Methods

In the paper, we analyze EDEN using the theoretical framework of non-linear dynamical systems and some new tools obtained by extending the concept of energy functions to the temporal case. The simulations were coded in Python and run in the Unity supercomputing cluster. The github repo for running the capacity experiments can be found at https://github.com/arjunkaruvally/EDEN_torch.

### A.1 Simulations

For all numerical simulations of network state dynamics, we used the Euler integration procedure with a step size of $0.01$. The memories in EDEN are defined as random binary vectors with each dimension of the memory in the model drawn from the Rademacher distribution $\Pr\left[\xi_j^{(\mu)} = +1\right] = \Pr\left[\xi_j^{(\mu)} = -1\right] = 1/2$. The similarity between the population activity and each memory is evaluated using the average overlap (Mattis magnetization) of the neural activity with each of the stored memories, defined as $m_\mu^x = \frac{1}{N} \sum_{j=1, j \neq i}^{N} \xi_j^{(\mu)} x_j$ where $\xi_j^{(\mu)}$ is the $\mu^{th}$ memory in the system and $N$ is the number of feature neurons. $x$ can be either the state of the feature neurons or the slow population. These memories are organized as long cyclic sequence episodes: $\xi^{(1)} \to \xi^{(2)} \to \dots \xi^{(P)} \to \xi^{(1)}$. The input cue to the system is the memory $\xi^{(1)}$, which is initialized as the feature layer state. The slow population is initialized to 0.

For Figure 2, 3, 5, the simulations were run with $N = 100, \alpha_s = 0.98, \alpha_c = 1.0, \mathcal{T}_f = 1.0$, and $\mathcal{T}_d = 20.0$. The code for the simulations is available in the repository: anonymous repo

#### A.1.1 Fixed point analysis

We used a fixed point finding algorithm to find the fixed points of the energy surface for the fast population at each time step [44]. The algorithm uses an iterative process to find the fixed points of the energy surface evaluated from a given position on the energy surface. Starting from the neuron state on the energy landscape, the state is updated to follow the direction of the energy gradient till no more updates are possible, indicating convergence to a fixed point on the energy surface.

### A.2 Capacity Experiments

To evaluate capacity, we ran simulations to estimate the probability of errors in retrieving single bits $\left( \Pr\left[ v_i(t_e) \cdot \xi_i^{(\mu)} \geq 1 - \epsilon \right] \right)$ for the fixed point error rate $\epsilon = 10^{-3}$. For each neuron setting $N \in \{10, 12, \dots 32\}$ and the number of memories from $P \in \{1, 2 \dots 2^N\}$, the probability is estimated using Monte Carlo simulations. 100 seeds of memory initializations were taken with the memories sampled without replacement to avoid confusion in the retrieved memory sequence. After evaluating the single-bit error probability, the maximum number of memories to be stored is computed for an error rate of $\delta = 10^{-3}$. The precise setting of $\epsilon$ and $\delta$ contribute only linearly to the exponential capacity [45, 37].

## B Energy Function Dynamics

The introduction of asymmetric synapses to the symmetric Hopfield network means that the standard energy minimization argument does not hold for EDEN. However, we find here that under sufficiently slow-changing asymmetric interactions the energy argument is valid in short-timescales. To illustrate this, we analyze the derivative of the energy function with respect to time to uncover how the energy function behaves along the dynamic trajectory of the system

$$\frac{dE}{dt} = \sum_i v_i \frac{dv_i}{dt} - \frac{1}{\alpha_s} \sum_{i,\mu} \left( \frac{z_\mu}{\sum_\nu z_\nu} \left( \alpha_s \xi_i^{(\mu)} \frac{dv_i}{dt} + \alpha_c \xi_i^{(\mu-1)} \frac{ds_i}{dt} \right) \right) \tag{12}$$

$$\frac{dE}{dt} = \sum_i v_i \frac{dv_i}{dt} - \sum_{i,\mu} \frac{z_\mu}{\sum_\nu z_\nu} \xi_i^{(\mu)} \frac{dv_i}{dt} + \frac{\alpha_c}{\alpha_s} \sum_{i,\mu} \frac{z_\mu}{\sum_\kappa z_\kappa} \xi_i^{(\mu-1)} \frac{ds_i}{dt} \tag{13}$$

$$\frac{dE}{dt} = \sum_i \frac{dv_i}{dt} \left( v_i - \sum_\mu \frac{z_\mu}{\sum_\nu z_\nu} \xi_i^{(\mu)} \right) + \frac{\alpha_c}{\alpha_s} \sum_{i,\mu} \frac{z_\mu}{\sum_\kappa z_\kappa} \xi_i^{(\mu-1)} \frac{ds_i}{dt} \tag{14}$$

$$\frac{\mathrm{d}E}{\mathrm{d}t} = -\sum_i \mathcal{T}_f \left(\frac{\mathrm{d}v_i}{\mathrm{d}t}\right)^2 + \frac{\alpha_c}{\alpha_s} \sum_{i,\mu} \frac{z_\mu}{\sum_\kappa z_\kappa} \xi_i^{(\mu-1)} \frac{\mathrm{d}s_i}{\mathrm{d}t} \tag{15}$$

The energy function dynamics splits into two terms - one term, which is always negative (analogous to the case of standard Hopfield networks), and the other term, which depends on the rate of change of the slow signal. In the adiabatic limit of the slow signal, the negative term dominates and the network dynamics always converge on the energy surface.

## C  Slow Population Dynamics

The slow population dynamics is a linear ODE, which can be solved exactly analytically under the fast $v_i$ assumptions

$$\mathcal{T}_d \frac{\mathrm{d}s_i}{\mathrm{d}t} = v_i - s_i \tag{16}$$

$$\mathrm{d}s_i + \frac{1}{\mathcal{T}_d} s_i \, \mathrm{d}t = \frac{1}{\mathcal{T}_d} v_i \, \mathrm{d}t \tag{17}$$

Use integrating factor $\exp\left(\frac{t}{\mathcal{T}_d}\right)$

$$\exp\left(\frac{t}{\mathcal{T}_d}\right) \mathrm{d}s_i + \frac{1}{\mathcal{T}_d} \exp\left(\frac{t}{\mathcal{T}_d}\right) s_i \, \mathrm{d}t = \frac{1}{\mathcal{T}_d} \exp\left(\frac{t}{\mathcal{T}_d}\right) v_i \, \mathrm{d}t \tag{18}$$

$$\mathrm{d}\left(s_i \exp\left(\frac{t}{\mathcal{T}_d}\right)\right) = \frac{1}{\mathcal{T}_d} \exp\left(\frac{t}{\mathcal{T}_d}\right) v_i \, \mathrm{d}t \tag{19}$$

Integrate both sides

$$\int_{t_0}^t \mathrm{d}\left(s_i \exp\left(\frac{t}{\mathcal{T}_d}\right)\right) = \frac{1}{\mathcal{T}_d} \int_{t_0}^t \exp\left(\frac{s}{\mathcal{T}_d}\right) v_i(s) \, \mathrm{d}s \tag{20}$$

$$\left[s_i \exp\left(\frac{t}{\mathcal{T}_d}\right)\right]_{t_0}^t = \frac{1}{\mathcal{T}_d} \int_{t_0}^t \exp\left(\frac{s}{\mathcal{T}_d}\right) v_i(s) \, \mathrm{d}s \tag{21}$$

$$s_i(t) \exp\left(\frac{t}{\mathcal{T}_d}\right) = s_i(t_0) \exp\left(\frac{t_0}{\mathcal{T}_d}\right) + \frac{1}{\mathcal{T}_d} \int_{t_0}^t \exp\left(\frac{s}{\mathcal{T}_d}\right) v_i(s) \, \mathrm{d}s \tag{22}$$

$$s_i(t) = s_i(t_0) \exp\left(\frac{t_0 - t}{\mathcal{T}_d}\right) + \frac{1}{\mathcal{T}_d} \int_{t_0}^t \exp\left(\frac{s - t}{\mathcal{T}_d}\right) v_i(s) \, \mathrm{d}s \tag{23}$$

Without the input signal $s$, the network is a continuous-time version of exponential static memory [21] and hence has the same capacity guarantees. For analytical simplicity, we assume circularly connected memories where $\xi^{(\mu-1)} \to \xi^{(\mu)}, \mu > 1$ and $\xi^{(P)} \to \xi^{(1)}$, where $P$ is the total number of memories. We assume that the transition is instantaneous in the slow timescale $\mathcal{T}_d$, and neglect the effect of transients in the slow population. Without any loss of generality, when the network state starts at state $\xi^{(2)}$, the slow population state has two components - the previous memory state $\xi^{(1)}$ and the current memory state $\xi^{(2)}$. We assume that $\mathcal{T}_d \gg \mathcal{T}_f$, so the transient states are negligible. $\lambda$ is a factor that controls to what extent the previous state is reflected in the slow population before the transition occurs. The $\lambda$ is computed analytically in Appendix D.

$$\boxed{s_i(t) = \lambda \, \xi_i^{(1)} \exp\left(-\frac{t}{\mathcal{T}_d}\right) + \xi_i^{(2)} \left(1 - \exp\left(-\frac{t}{\mathcal{T}_d}\right)\right)} \tag{24}$$

## D Escape Time

To ease the computation of the escape time in relation to the parameters of the network, we scale the timescale of the network dynamics by the substitution $t' = t\,\mathcal{T}_f$. This removes $\mathcal{T}_f$ from the dynamical equations and replaces its effect as the timescale ratio $\tau = \mathcal{T}_d/\mathcal{T}_f$. The slow population dynamics for the rescaled system is

$$\frac{\mathcal{T}_d}{\mathcal{T}_f}\frac{\mathrm{d}s_i}{\mathrm{d}t} = v_i - s_i \tag{25}$$

and has the following analytic form for the trajectory.

$$s_i(t) = \lambda\,\xi_i^{(1)}\exp\left(-\frac{t}{\tau}\right) + \xi_i^{(2)}\left(1 - \exp\left(-\frac{t}{\tau}\right)\right) \tag{26}$$

To compute average escape time, we consider the two memory contributions $C_2, C_3$ on the energy function, for the sequence transition $\xi^{(1)} \to \xi^{(2)} \to \xi^{(3)}$ and analyze for the transition $\xi^{(2)} \to \xi^{(3)}$. That is, $v_i = \xi_i^{(2)}$ and $s_i(t) = \lambda\,\xi_i^{(1)}\exp\left(-\frac{t}{\tau}\right) + \xi_i^{(2)}\left(1 - \exp\left(-\frac{t}{\tau}\right)\right)$, where $\lambda$ is the coefficient of the contribution of $\xi_i^{(1)}$ to the delayed state before transition to $\xi^{(\mu)}i2$

$$C_2 + C_3 = \exp\left(\alpha_s \sum_i \xi_i^{(2)} v_i + \alpha_c \sum_i \xi_i^{(1)} s_i\right) + \exp\left(\alpha_s \sum_i \xi_i^{(3)} v_i + \alpha_c \sum_i \xi_i^{(2)} s_i\right) \tag{27}$$

Substituting $v_i = \xi_i^{(2)}$ and $s_i(t) = \lambda\,\xi_i^{(1)}\exp\left(-\frac{t}{\tau}\right) + \xi_i^{(2)}\left(1 - \exp\left(-\frac{t}{\tau}\right)\right)$

$$C_2 + C_3$$
$$= \exp\left(\alpha_s \sum_i \xi_i^{(2)}\xi_i^{(2)} + \lambda\,\alpha_c \sum_i \xi_i^{(1)}\xi_i^{(1)}\exp\left(-\frac{t}{\tau}\right) + \alpha_c \sum_i \xi_i^{(1)}\xi_i^{(2)}\left(1 - \exp\left(-\frac{t}{\tau}\right)\right)\right)$$
$$+ \exp\left(\alpha_s \sum_i \xi_i^{(3)}\xi_i^{(2)} + \lambda\,\alpha_c \sum_i \xi_i^{(2)}\xi_i^{(1)}\exp\left(-\frac{t}{\tau}\right) + \alpha_c \sum_i \xi_i^{(2)}\xi_i^{(2)}\left(1 - \exp\left(-\frac{t}{\tau}\right)\right)\right) \tag{28}$$

The energy minima is characterized by the competition between the two memory contributions. Now, we take the ansatz that the transition occurs when the energy contribution to the minima $C_2 < C_2$. Since $\exp$ is a monotonic function, this can be written as

$$\alpha_s \sum_i \xi_i^{(2)}\xi_i^{(2)} + \lambda\,\alpha_c \sum_i \xi_i^{(1)}\xi_i^{(1)}\exp\left(-\frac{t}{\tau}\right) + \alpha_c \sum_i \xi_i^{(1)}\xi_i^{(2)}\left(1 - \exp\left(-\frac{t}{\tau}\right)\right)$$
$$< \alpha_s \sum_i \xi_i^{(3)}\xi_i^{(2)} + \lambda\,\alpha_c \sum_i \xi_i^{(2)}\xi_i^{(1)}\exp\left(-\frac{t}{\tau}\right) + \alpha_c \sum_i \xi_i^{(2)}\xi_i^{(2)}\left(1 - \exp\left(-\frac{t}{\tau}\right)\right) \tag{29}$$

At the large $N$ limit, the terms $\sum_i \xi_i^{(\mu)}\xi_i^{(\mu)} \sim \mathcal{N}(0,\sigma)$ for $\mu \neq \nu$ can be approximated by a normal distributed random variable. Let $\epsilon_i \sim \mathcal{N}(0,\sigma_i)$ and $\epsilon_1 = \sum_i \xi_i^{(1)}\xi_i^{(2)}, \epsilon_2 = \sum_i \xi_i^{(3)}\xi_i^{(2)}, \epsilon_3 = \sum_i \xi_i^{(2)}\xi_i^{(1)}$

$$\alpha_s N + \lambda\,\alpha_c N \exp\left(-\frac{t}{\tau}\right) + \alpha_c \epsilon_1\left(1 - \exp\left(-\frac{t}{\tau}\right)\right) < \alpha_s \epsilon_2 + \lambda\,\alpha_c \epsilon_3 + \alpha_c N\left(1 - \exp\left(-\frac{t}{\tau}\right)\right) \tag{30}$$

$$\alpha_s + \lambda\,\alpha_c \exp\left(-\frac{t}{\tau}\right) + \alpha_c \frac{\epsilon_1}{N}\left(1 - \exp\left(-\frac{t}{\tau}\right)\right) < \alpha_s \frac{\epsilon_2}{N} + \lambda\,\alpha_c \frac{\epsilon_3}{N} + \alpha_c\left(1 - \exp\left(-\frac{t}{\tau}\right)\right) \tag{31}$$

$$\exp\left(-\frac{t}{\tau}\right)\left(\lambda+1-\frac{\epsilon_1}{N}\right)\alpha_c < \alpha_s\left(\frac{\epsilon_2}{N}-1\right)+\alpha_c\left(1+\lambda\frac{\epsilon_3}{N}-\frac{\epsilon_1}{N}\right) \tag{32}$$

Let $r = \frac{\alpha_s}{\alpha_c}$

$$\exp\left(-\frac{t}{\tau}\right)\left(\lambda+1-\frac{\epsilon_1}{N}\right) < r\left(\frac{\epsilon_2}{N}-1\right)+\left(1+\lambda\frac{\epsilon_3}{N}-\frac{\epsilon_1}{N}\right) \tag{33}$$

$$\exp\left(\frac{t}{\tau}\right) > \frac{\left(\lambda+1-\frac{\epsilon_1}{N}\right)}{r\left(\frac{\epsilon_2}{N}-1\right)+\left(1+\lambda\frac{\epsilon_3}{N}-\frac{\epsilon_1}{N}\right)} \tag{34}$$

Applying $\ln$ function on both sides

$$t > \tau\left[\ln\left(\lambda+1-\frac{\epsilon_1}{N}\right)-\ln\left(r\left(\frac{\epsilon_2}{N}-1\right)+\left(1+\lambda\frac{\epsilon_3}{N}-\frac{\epsilon_1}{N}\right)\right)\right] \tag{35}$$

The time to escape is written as a random variable

$$t_e = \tau\left[\ln\left(\lambda+1-\frac{\epsilon_1}{N}\right)-\ln\left(r\left(\frac{\epsilon_2}{N}-1\right)+\left(1+\lambda\frac{\epsilon_3}{N}-\frac{\epsilon_1}{N}\right)\right)\right] \tag{36}$$

For large $N$, the expected escape time using the delta method to approximate the log of random variable as normal distributed is obtained as

$$\langle t_e \rangle = \tau\left[\ln(\lambda+1)-\ln(1-r)\right] \tag{37}$$

Now that we have the time to escape, we compute the slow signal $s$ at the transition point:

$$s_i(t) = \lambda\xi_i^{(1)}\exp\left(-\frac{t_e}{\tau}\right)+\xi_i^{(2)}\left(1-\exp\left(-\frac{t_e}{\tau}\right)\right) \tag{38}$$

Substituting the equations for escape times,

$$s_i(t) = \lambda\xi_i^{(1)}\left(\frac{1-r}{\lambda+1}\right)+\xi_i^{(2)}\left(1-\left(\frac{1-r}{\lambda+1}\right)\right) \tag{39}$$

$$s_i(t) = \lambda\xi_i^{(1)}\left(\frac{1-r}{\lambda+1}\right)+\xi_i^{(2)}\left(\frac{\lambda+r}{\lambda+1}\right) \tag{40}$$

At transition,

$$\frac{\lambda+r}{\lambda+1} = \lambda \tag{41}$$

$$\boxed{\lambda = \sqrt{\frac{\alpha_s}{\alpha_c}}} \tag{42}$$

Therefore, before transition, the delay signal will be

$$s_i(t) = \sqrt{r}\xi_i^{(2)}+\xi_i^{(1)}(1-\sqrt{r}) \tag{43}$$

This computation of $\lambda$ seem to generate confusions. So, we have decided to provide a detailed reasoning. For a transition $\xi^{(1)} \to \xi^{(2)}$, $\lambda$ is a factor quantifying the extend to which the previous

state $\xi^{(1)}$ is present in the slow population when the state transition occurs. Using the Markovian assumption, we assume that $\xi^{(P)}$ is negligible in the slow population when the transition occurred". We perform our analysis just after the transition $\xi^{(1)} \to \xi^{(2)}$ happened where the "old" pattern is indeed $\xi^{(1)}$. The escape time we compute is for the state transition $\xi^{(2)} \to \xi^{(3)}$. Now, the definition of $\lambda$ is used again as before but on the transition $\xi^{(2)} \to \xi^{(3)}$ similar to above, except now the "old" pattern is $\xi^{(2)}$.

An alternate way to think about $\lambda$ is by imagining a factor corresponding to a memory in the slow variable that increases as the network stays in a meta-stable memory state. Now, this factor ideally would reach 1 asymptotically over time, while any "old information" exponentially decays to 0 at which point the fast variable escapes the memory state. Instead of exactly 1, we use a factor $\lambda$ and compute what this is based on the parameters we have in our model. A sanity check is to verify if the factor at escape time in the most ideal Markovian case is very close to 1, which we indeed find in our analysis. For perfect sequential transitions, $\sqrt{r} \to 1$. This guarantees that the old memory is completely lost when the transition occurs and the accurate next state is retrieved. Now, to compute the analytical escape time:

$$\langle t_e \rangle = \tau \left[ \ln\left(\sqrt{r} + 1\right) - \ln(1 - r) \right] \tag{44}$$

$$\langle t_e \rangle = \tau \left[ \ln\left(\sqrt{r} + 1\right) - \ln(1 - r) \right] \tag{45}$$

$$\boxed{\langle t_e \rangle = -\frac{\mathcal{T}_d}{\mathcal{T}_f} \ln\left(1 - \sqrt{\frac{\alpha_s}{\alpha_c}}\right)} \tag{46}$$

## E  Capacity

There is a rich literature analyzing the capacity of energy-based networks like Hopfield networks. The capacity is defined as the scaling relationship between the number of dimensions in the state space of the network (the number of feature neurons) and the maximum number of memories that can be stored. It is typical to assume that minor errors are allowed as long as the error does not scale with the number of neurons. We follow the analysis introduced by Petritis [45] and recently used in [37]. Recall that capacity is defined as the maximum number of memories that can be stored such that each dimension of the fixed point encounters an error of $\epsilon$ with a probability $\delta$. Mathematically,

$$C(N, \epsilon, \delta) = \max\left\{ P \in \mathbb{N} : \Pr\left[ v_i(t_e) \cdot \xi_i^{(\mu)} \geq 1 - \epsilon \right] \geq 1 - \delta \right\} \tag{47}$$

Typically, $v_i(t_e)$ requires solving a system of non-linear dynamical equations. Since we have access to the analytic energy function of the system, we compute the fixed point of the energy function at $t_e$ and use it as the proxy for the network state at that time.

### E.1  Reference Network

The reference network is defined by the following equations:

$$\begin{cases} \mathcal{T}_f \dfrac{dv_i}{dt} &= \alpha_s \sum_{\mu j} \xi_{i\mu}\, \xi_{j\mu}\, \sigma(v_i) + \alpha_c \sum_{\mu,j} \xi_{i\mu}\, \xi_{j\mu-1}\, s_j - v_i\,, \\[2mm] \mathcal{T}_d \dfrac{ds_i}{dt} &= v_i - s_i\,. \end{cases} \tag{48}$$

The energy function for this network is given as:

$$E_{\text{ref}}(v) = \frac{\sum_i v_i^2}{2} - \frac{1}{2\alpha_s} \sum_\mu (\alpha_s \langle \xi^{(\mu)}, \sigma(v) \rangle + \alpha_c \langle \xi^{(\mu-1)}, s \rangle)^2 \tag{49}$$

Without loss of generality, the fixed point of the energy surface at the point of transition $\xi^{(2)} \to \xi^{(3)}$ is given by

$$v_i^* = \alpha_s \sum_{\mu,j} \xi_i^{(\mu)} \xi_j^{(\mu)} v_j^* + \alpha_c \sum_{\mu,j} \xi_i^{(\mu)} \xi_j^{(\mu-1)} s_j(t_e)$$

$$v_i^* = \alpha_s \sum_{\mu j} \xi_i^{(\mu)} \xi_j^{(\mu)} v_j^* + \alpha_c \sum_{\mu,j} \xi_i^{(\mu)} \xi_j^{(\mu-1)} \xi_j^{(2)}$$

We then quantify the probability for the failure of a single bit by computing the following probability, where $v_i(t_e) = v_i^*$:

$$\Pr\left[v_i(t_e) \cdot \xi_i^{(3)} < 1 - \epsilon\right]$$

$$v_i^* \cdot \xi_i^{(3)} = \alpha \left[2(N-1) + \sum_{\mu \neq 3} \xi_i^{(\mu)} \langle \xi^{(\mu)}, v_i^* \rangle + \sum_{\mu \neq 3} \xi_i^{(\mu)} \langle \xi^{(\mu-1)}, \xi^{(2)} \rangle\right] \tag{50}$$

Let $\alpha = \frac{1}{2(N-1)}$ to simplify the effect of the discontinuity

$$= 1 + \frac{1}{2(N-1)} \sum_{\mu \neq 3} \xi_i^{(\mu)} \xi_i^{(3)} \left(\langle \xi^{(\mu)}, \xi^{(3)} \rangle + \langle \xi^{(\mu-1)}, \xi^{(2)} \rangle\right)$$

Introduce the random variable $\chi$

$$\chi = \frac{1}{2(N-1)} \sum_{\mu \neq 3} \xi_i^{(\mu)} \xi_i^{(3)} \left(\langle \xi^{(\mu)}, \xi^{(3)} \rangle + \langle \xi^{(\mu-1)}, \xi^{(2)} \rangle\right)$$

Since $\xi_i^{(\mu)}$'s are Rademacher distributed, the r.v can be simplified as

$$\chi = \frac{1}{2(N-1)} \sum_{i=1} \left(\sum_{\mu \neq 3} R_i^{(\mu)} + \sum_{\nu \neq 3} R_i^{(\nu)}\right)$$

Here, $R_i^{(\mu)}, R_i^{(\nu)}$ are Rademacher distributed random variables. The probability of single bit failure is reformulated in the new random variable as:

$$\Pr[\,|\chi| \geq \epsilon\,]$$

The moments of $\chi$ is then computed to find the bounds on the failure probability.

### E.1.1  Moments

**First Moment (Mean)** Note the the distribution is symmetric around the origin, which gives the first moment as

$$\mathbb{E}[\chi] = 0$$

**4.2 Second Moment (Variance)**

$$\mathbb{V}[\chi] = \frac{(N-1)}{4(N-1)^2} \mathbb{V}\left[\sum_\mu R_i^{(\mu)} + \sum_{\nu \neq 3} R_i^{(\nu)}\right]$$

$$\mathbb{V}[\chi] = \frac{1}{4(N-1)} \mathbb{V}\left[\sum_\mu R_i^{(\mu)} + \sum_{\nu \neq 3} R_i^{(\nu)}\right]$$

$$\mathbb{V}[\chi] = \frac{2(P-1)}{4(N-1)} \mathbb{V}\left[R_i^{(\mu)}\right]$$

$$\mathbb{V}[\chi] = \frac{(P-1)}{2(N-1)}$$

**Bounds of chi**

Chebyshev's inequality

$$\Pr[|\chi| \geq \epsilon] \leq \frac{\mathbb{V}[\chi]}{\epsilon^2}$$

$$\Pr[|\chi| \geq \epsilon] \leq \frac{(P-1)}{2(N-1)\,\epsilon^2}$$

Using our definition of capacity, we obtain

$$\frac{(P-1)}{2(N-1)\,\epsilon^2} = \delta$$

Solving for $P$, we obtain

$$P = 1 + 2\delta\epsilon^2(N-1)$$

which is linear in the number of neurons. For constant error rates, $\epsilon$ and $\delta$, the capacity has an asymptotic scaling of $O(N)$ in line with prior classical Hopfield Network bounds.

### E.2 EDEN

We follow a similar approach for EDEN and set $\alpha_c = r\alpha_s = r\alpha$. The fixed point of EDEN is given as

$$v_i^* = \sum_\mu \xi_i^{(\mu)} \sigma(\alpha(r\langle \xi^{(\mu)}, v^* \rangle + \langle \xi^{(\mu-1)}, \xi^{(2)} \rangle)))$$

Let $Z = \sum_{\nu \neq 3} \frac{\exp\left(\alpha(r\langle \xi^{(\mu)}, \xi^{(3)} \rangle + \langle \xi^{(\mu-1)}, \xi^{(2)} \rangle))\right)}{\exp(\alpha(1+r)(N-1))}$

$$1 - v_i^*(t_e)\,\xi_i^{(3)} = \frac{Z}{1+Z} - \sum_{\mu \neq 3} \xi_i^{(\mu)} \xi_i^{(3)} \frac{\exp\left(\alpha(r\langle \xi^{(\mu)}, \xi^{(3)} \rangle + \langle \xi^{(\mu-1)}, \xi^{(2)} \rangle - (r+1)(N-1))\right)}{1+Z}$$

$$(51)$$

There are two random variables in the quantity of interest. The first $Z$ is a sum of many terms, and we replace the sum with its mean for easier computation. The mean field approximation becomes valid in large $P$ limits which we consider in the paper.

$$Z = \frac{\sum_{\nu \neq 3} \prod_{j \neq i} \exp\left(\alpha r\, \xi_j^{(\nu)} \xi_j^{(3)}\right) \exp\left(\alpha\, \xi_j^{(\nu-1)} \xi_j^{(2)}\right)}{\exp(\alpha(r+1)(N-1)))}$$

introduce an r.v $x_j^{(\mu)} = \xi_j^{(\mu)} \xi_j^{(3)} \sim$ Rademacher and $y_j^{(\mu)} = \xi_j^{(\mu)} \xi_j^{(2)} \sim$ Rademacher

$$Z = \frac{\sum_{\nu \neq 3} \prod_{j \neq i} \exp\left(\alpha r\, x_j^{(\mu)}\right) \exp\left(\alpha\, y_j^{(\mu)}\right)}{\exp(\alpha(r+1)(N-1)))}$$

$$\mathbb{E}[Z] = \frac{\left(\mathbb{E}\left[\exp\left(\alpha r x_j^{(\mu)}\right)\right] \mathbb{E}\left[\exp\left(\alpha y_j^{(\mu)}\right)\right]\right)^{(N-1)}}{\exp(\alpha(r+1)(N-1)))}$$

$$\mathbb{E}[Z] = (P-1)\left(\frac{\cosh(\alpha r)}{\exp(r\alpha)} \frac{\cosh(\alpha)}{\exp(\alpha)}\right)^{(N-1)}$$

The $Z$ is then replaced with the mean value. Also define a new parameter $\beta_x = \frac{\cosh(x)}{\exp(x)}$

$$\chi = 1 - v_i^*(t_e)\,\xi_i^{(3)} = \frac{(P-1)(\beta_{\alpha r}\beta_\alpha)^{(N-1)} - \sum_{\mu \neq 3} \xi_i^{(\mu)} \xi_i^{(3)} \prod_{j \neq i} \frac{\exp\left(\alpha r\, \xi_j^{(\nu)} \xi_j^{(3)}\right)}{\exp(\alpha r(N-1))} \frac{\exp\left(\alpha\, \xi_j^{(\nu-1)} \xi_j^{(2)}\right)}{\exp(\alpha(N-1))}}{1 + (P-1)(\beta_{\alpha r}\beta_\alpha)^{(N-1)}}$$

$$(52)$$

We compute the expectation and variance to characterize the distribution of $\chi$. When computing the expectation, the second term does not contribute to the expectation due to the symmetry of the distribution.

$$\mathbb{E}[\chi] = \frac{(P-1)(\beta_{\alpha r}\beta_\alpha)^{(N-1)}}{1 + (P-1)(\beta_{\alpha r}\beta_\alpha)^{(N-1)}}$$

The independence of dimensions and memories guarantees that the covariance is 0 for the second term, resulting in the variance.

$$\mathbb{V}[\chi] = \frac{(P-1)(\beta_{2\alpha r}\beta_{2\alpha})^{(N-1)}}{(1 + (P-1)(\beta_{\alpha r}\beta_\alpha)^{(N-1)})^2}$$

The general distribution of $\chi$ is complicated, but it is symmetric around its mean. We, therefore, use moment matching to approximate the distribution of $\chi$ using Gaussian distribution.

$$\mathbb{E}[\chi] = \mu = \frac{(P-1)(\beta_{\alpha r}\beta_\alpha)^{(N-1)}}{1 + (P-1)(\beta_{\alpha r}\beta_\alpha)^{(N-1)}}$$

$$\mathbb{V}[\chi] = \sigma^2$$

$$\sigma = \frac{\sqrt{(P-1)}\,(\beta_{2\alpha r}\beta_{2\alpha})^{\frac{(N-1)}{2}}}{1 + (P-1)(\beta_{\alpha r}\beta_\alpha)^{(N-1)}}$$

$$\chi \sim \mathcal{N}(\mu, \sigma^2)$$

$$\Pr[\chi \le \epsilon] = \Phi(\frac{\epsilon - \mu}{\sigma}) = 1 - \delta$$

Here, $\Phi$ is the Gaussian CDF which does not have a closed-form expression, but it can be approximated analytically by

$$\Phi(\frac{\epsilon - \mu}{\sigma}) \approx \frac{\exp(2kx)}{(1 + \exp(2kx))} \quad k = \sqrt{\frac{2}{\pi}} \quad x = \frac{\epsilon - \mu}{\sigma}.$$

### E.2.1 In the large $N$ limit, $\delta \to 0$

For a given error tolerance $\epsilon > 0$, the success rate (given by $1 - \delta$) approaches 1.

$$1 - \delta = \left[1 + \exp\left(2k\left(\frac{P(\beta_{\alpha r}\beta_\alpha)^{(N-1)}(\epsilon - 1) + \epsilon}{\sqrt{P}(\beta_{2\alpha r}\beta_{2\alpha})^{(N-1)/2}}\right)\right)\right]^{-1}$$

Using the property that $\epsilon \ll 1$,

$$= \left[1 + \exp\left(2k\left(\frac{-P(\beta_{\alpha r}\beta_\alpha)^{(N-1)} + \epsilon}{\sqrt{P}(\beta_{2\alpha r}\beta_{2\alpha})^{(N-1)/2}}\right)\right)\right]^{-1}$$

$$= \left[1 + \exp\left(-2k\sqrt{P}\left(\frac{\cosh(\alpha r)\cosh(\alpha)}{\sqrt{\cosh(2\alpha r)\cosh(2\alpha)}}\right)^{(N-1)}\right)\right.$$

$$\left. \exp\left(2k\frac{\epsilon}{\sqrt{P}}(\beta_{2\alpha r}\beta_{2\alpha})^{-(N-1)/2}\right)\right]^{-1} \tag{53}$$

Now, taking the limit $N \to \infty$ since $\alpha r, \alpha > 0$,

$$(\beta_{2\alpha r}\beta_{2\alpha})^{-1} > 1$$

and $\beta_{2\alpha r}\beta_{2\alpha} \to \infty$ when $N \to \infty$

$$= \left[1 + \exp\left(-2k\sqrt{P}\left(\frac{\cosh(\alpha r)\cosh(\alpha)}{\sqrt{\cosh(2\alpha r)\cosh(2\alpha)}}\right)^{(N-1)}\right)\right]^{-1}$$

also, $\frac{\cosh(\alpha r)\cosh(\alpha)}{\sqrt{\cosh(2\alpha r)\cosh(2\alpha)}} > 1, \forall \alpha, r > 0$ so taking $N \to \infty$ gives

$$\delta = 0$$

Q.E.D

### E.2.2 EDEN has exponential capacity

$a = \sqrt{(\beta_{2\alpha r} \beta_{2\alpha})}$ and $b = \frac{\beta_{\alpha r} \beta_\alpha}{\sqrt{(\beta_{2\alpha r} \beta_{2\alpha})}}$

$$\frac{b}{a} = \frac{\beta_{\alpha r} \beta_\alpha}{\beta_{2\alpha r} \beta_{2\alpha}}$$

$$\exp\left(-2k\sqrt{P}\left(\frac{\beta_{\alpha r}\beta_\alpha}{\sqrt{\beta_{2\alpha r}\beta_{2\alpha}}}\right)^{(N-1)}\right) \exp\left(2k\frac{\epsilon}{\sqrt{P}}\left(\beta_{2\alpha r}\beta_{2\alpha}\right)^{-(N-1)/2}\right) = \frac{1-\delta}{\delta} \tag{54}$$

$$-2k\sqrt{P}\left(\frac{\beta_{\alpha r}\beta_\alpha}{\sqrt{\beta_{2\alpha r}\beta_{2\alpha}}}\right)^{(N-1)} + 2k\frac{\epsilon}{\sqrt{P}}\left(\beta_{2\alpha r}\beta_{2\alpha}\right)^{-(N-1)/2} = \ln\left(\frac{1-\delta}{\delta}\right) \tag{55}$$

$$-2k\sqrt{P}\left(\frac{\beta_{\alpha r}\beta_\alpha}{\sqrt{\beta_{2\alpha r}\beta_{2\alpha}}}\right)^{(N-1)} + 2k\frac{\epsilon}{\sqrt{P}}\left(\beta_{2\alpha r}\beta_{2\alpha}\right)^{-(N-1)/2} = \ln\left(\frac{1-\delta}{\delta}\right) \tag{56}$$

$$P + \sqrt{P}\frac{1}{2k}\ln\left(\frac{1-\delta}{\delta}\right)\left(\frac{\sqrt{\beta_{2\alpha r}\beta_{2\alpha}}}{\beta_{\alpha r}\beta_\alpha}\right)^{(N-1)} - \frac{\epsilon}{(\beta_{\alpha r}\beta_\alpha)^{(N-1)}} = 0 \tag{57}$$

which is a quadratic equation in $\sqrt{P}$ and can be solved to obtain

$$\sqrt{P} = \frac{1}{2k}\ln\left(\frac{\delta}{1-\delta}\right)\left(\frac{\sqrt{(\beta_{2\alpha r}\beta_{2\alpha})}}{\beta_{\alpha r}\beta_\alpha}\right)^{N-1}$$

$$+ \sqrt{\left[\frac{\ln\left(\frac{1-\delta}{\delta}\right)}{2k}\right]^2 \left[\frac{\sqrt{(\beta_{2\alpha r}\beta_{2\alpha})}}{\beta_{\alpha r}\beta_\alpha}\right]^{2(N-1)} + \frac{4\epsilon}{(\beta_{\alpha r}\beta_\alpha)^{N-1}}} \tag{58}$$

Let $c = \left(\frac{\beta_{\alpha r}\beta_\alpha}{\beta_{2\alpha r}\beta_{2\alpha}}\right)^{(N-1)}$

$$P = \left(\frac{1}{\beta_{\alpha r}\beta_\alpha}\right)^{N-1}\left(\frac{1}{2k\sqrt{c}}\ln\left(\frac{\delta}{1-\delta}\right) + \sqrt{\frac{1}{c}\left[\frac{\ln\left(\frac{1-\delta}{\delta}\right)}{2k}\right]^2 + 4\epsilon}\right) \tag{59}$$

