# OpenReview forum: "Exponential Dynamic Energy Network for High Capacity Sequence Memory"
_NeurIPS.cc/2025/Conference — NeurIPS 2025 poster_

### Official Review · Reviewer_hmVt · 2025-06-29

**Clarity:** 3
**Significance:** 3
**Originality:** 2
**Rating:** 4
**Confidence:** 4

**Summary:**

This paper combines dense associative memory with sequential associative memory by introducing a time-scale-separated slow hidden population that controls the faster dense associative memory network. The authors show that the network can perform sequential memory retrieval with a capacity that grows exponentially with the number of visible neurons N. They analytically calculate the escape time and the memory capacity of the network and corroborate the theory with numerical simulations.

**Questions:**

The idea of using a slow population to control memory transitions was reported earlier in the literature, as the authors note, but they seem to have omitted another recent addition to the family [1]. In fact, the dynamics considered in Eq. (1) can be viewed as a soft-max generalization of the quadratic interaction reported in earlier literature. However, the present contribution does show that such a sequence Hopfield network can achieve an exponential memory capacity, which is novel.

The average escape time calculated in Eq. (5), while a good first approximation (as evidenced by the good match in Figure 3), does not consider the more nuanced kinetic effects that are important in energy-based models. In such non-equilibrium systems, the external drive (modeled here as the slow population) modifies the effective energy landscape, but the time it takes the state to escape a local energy minimum is generally different from the time it takes for the energy landscape itself to change (captured in the present calculation). Such a first-passage-time calculation is known as the Kramers problem, and its solution is approximately given by the Arrhenius law. For example, a similar first-passage-time calculation for a neural population under external drive was considered in [2]. Incorporating these kinetic considerations into the calculation would be an interesting next step.

Another issue is that although the capacity is exponential in N, the model requires one hidden unit per memory, so the total number of hidden units also grows exponentially with N. Consequently, when counting all neurons in the network, the overall capacity scaling is only linear. Having exponential capacity only in the visible neurons is therefore not sufficient; the authors should either devise a way to reduce the number of hidden units required or clarify this important point about scaling. Otherwise, the defined capacity is not the most relevant metric.

Finally, regarding Figure 5: to achieve exponential capacity the model assumes high-order, all-body interactions (obtained by expanding the exponential in the soft-max interaction term). While the figure nicely demonstrates the dynamical traces of the slow neurons, it is a bit of a stretch to call them time cells and ramping cells.

Refs:

[1] Herron, L., Sartori, P., & Xue, B. (2023). Robust retrieval of dynamic sequences through interaction modulation. PRX Life, 1(2), 023012.

[2] Zhong, W., Lu, Z., Schwab, D. J., & Murugan, A. (2020). Nonequilibrium statistical mechanics of continuous attractors. Neural computation, 32(6), 1033-1068.

**Ethical Concerns:**

["NO or VERY MINOR ethics concerns only"]

**Final Justification:**

This topic considered in this paper is not entirely novel: both the sequence memory task and the exponential interaction term (which leads to exponential capacity in the feature space) are already discussed extensively in the literature. However, I do appreciate the theoretical analysis which matches nicely with simulations. Therefore, I can only recommend borderline accept.

**Limitations:**

The authors did not include a discussion of the limitations.

**Quality:**

3

**Strengths And Weaknesses:**

Extending the much-discussed Hopfield network to perform sequential memory retrieval is an important problem. In this work, the authors conduct a detailed theoretical analysis and numerical experiments on this topic, and the results are well-presented. However, some of the analysis and claims need further improvement, as I elaborate below.

---

> ### Author Rebuttal · Authors · 2025-07-31
>
> We thank the reviewer for comments of our work and suggesting related work relevant to the paper. We also thank the reviewer for highlighting that the exponential capacity is novel in the context of sequential memories. We appreciate the thought of the reviewer to suggest incorporating kinetic effects to consider the dynamic properties of the network, which is required for more general settings of sequence models that generalize the Markovian sequence transition assumption we make in the paper.
>
> **Another issue is that although the capacity is exponential in N, the model requires one hidden unit per memory, so the total number of hidden units also grows exponentially with N. Consequently, when counting all neurons in the network, the overall capacity scaling is only linear. Having exponential capacity only in the visible neurons is therefore not sufficient; the authors should either devise a way to reduce the number of hidden units required or clarify this important point about scaling. Otherwise, the defined capacity is not the most relevant metric.**
>
> We thank the reviewer for the opportunity to clarify this point. The capacity argument is a common criticism of the literature of Dense Memory models and is used in many landmark paper. Here, there is a subtlety in the definition of capacity - i.e capacity is defined as the maximum number of memories that can be stored as function of the dimensions in the state space (Line 197 in the paper) and not the actual neurons in the network. This is a subtle point and we have adjusted our discussion to make this clear. We have also introduced a limitation section to enumerate this weakness in defining capacity in this manner.
>
> The reason why capacity is defined in this way is because the hidden neurons in Eqn (1) are often not explicitly written out as separate neurons, instead these variables are assumed to be part of dynamics of the visual neurons $(v_i)$ itself (by substituting $h_\mu$ in the dynamical equation of $v_i$). This requires an assumption that the synapses have many-body interactions (see reference \[1\] below). Even though this definition of capacity may seem to inflate the actual capacity, not all neural networks can accommodate arbitrary high hidden neurons for a fixed number of visual neurons, which is the main problem tackled by dense associative memory models. e.g. Classical Hopfield networks can accommodate only a small fraction of the original visual neurons as hidden neurons without introducing spurious patterns.
>
> \[1\] Krotov, Dmitry, and John J. Hopfield. "Large Associative Memory Problem in Neurobiology and Machine Learning." _International Conference on Learning Representations_. 2021
>
> **Finally, regarding Figure 5: to achieve exponential capacity the model assumes high-order, all-body interactions (obtained by expanding the exponential in the soft-max interaction term). While the figure nicely demonstrates the dynamical traces of the slow neurons, it is a bit of a stretch to call them time cells and ramping cells.**
>
> Thank you for raising this excellent point. The model we have is abstracted away from the biophysically realistic neural models like Hodgkin-Huxley neural networks. As such there are limitations to the extend to which the biological claims can be made. In the section, we make the case that the dynamical behavior is correlated - which is still interesting because this match to biology was not originally intended in the model. We only make the claim that the dynamical characteristics show similarities to biology. The perceived role of ramping/time cells also match some of the predictions we make from the theory. The evidence we show is only correlational and further work may be required to better quantify this connection

---

> > ### Comment · Reviewer_hmVt · 2025-08-05
> >
> > After reading the comments and rebuttals with Reviewer JBWE, I also agree that the term "Exponential" is a bit oversold. It is true that previous works use similar terminology, but that's all the reason not to put it in the title, in my opinion. Also, other reviewers pointed out the highly relevant reference Karuvally 2023, which after comparison I think the novelty in the current paper mainly lies in the theoretical analysis (which I appreciate), since the two key ingredients: slow-fast dynamics, exponential interaction term, are both not new. Therefore, I think the writing should emphasize that this is a theoretical analysis on Modern Hopfield Network with Sequence Memory, rather than emphasizing that this is a new architecture call EDEN.

---

> ### Author Response · Authors · 2025-08-05
>
> We thank the reviewer for engaging with our responses. We want to clarify that the term "Exponential" in the title refers to the exponential interaction terms in the model, which we indeed have in our model. i.e the hidden neurons have exponential activation with normalization in Eqn 1 of our paper.
>
> "Also, other reviewers pointed out the highly relevant reference Karuvally 2023, which after comparison I think the novelty in the current paper mainly lies in the theoretical analysis (which I appreciate), since the two key ingredients: slow-fast dynamics, exponential interaction term, are both not new. Therefore, I think the writing should emphasize that this is a theoretical analysis on Modern Hopfield Network with Sequence Memory, rather than emphasizing that this is a new architecture call EDEN."
>
> Thank you for pointing to the work and providing an opportunity to clarify the distinction with Karuvally 2023. **The model we introduce in Eqn 1 is different from Karuvally 2023**. Karuvally 2023 only considers polynomial interaction terms in their Dense GSEMM compared to the exponential terms we have in EDEN. We respectfully urge the reviewer to refer to the appropriate section of the Karuvally 2023 paper. The exponential terms are new and the theoretical analysis is a nontrivial extension of the Karuvally 2023 paper and provides rigor to the claims raised there.
>
> Karuvally, A., Sejnowski, T. &amp; Siegelmann, H.T.. (2023). General Sequential Episodic Memory Model. <i>Proceedings of the 40th International Conference on Machine Learning

---

> > ### Comment · Reviewer_JBWE · 2025-08-05
> >
> > >We want to clarify that the term "Exponential" in the title refers to the exponential interaction terms in the model
> >
> > This is a hilariously bold claim given this statement from the paper:
> >
> > > To develop *dynamic energy networks with exponential capacity*, we incorporated a slow-changing
> > signal that interacts asymmetrically with an *exponential capacity static energy network* introduced
> > in prior research

---

> > > ### Author Response · Authors · 2025-08-06
> > >
> > > We thank the reviewer for pointing to the part of the paper that caused the confusion regarding the teminology in the title. We propose the following revision to the phrasing
> > >
> > > "To develop a dynamic energy network with improved capacity, we incorporated a slow-changing signal that interacts asymmetrically with a modern Hopfield Network with exponential type interactions. The exponential interactions were shown to lead to exponential capacity (in the feature neurons) in prior static memory models (Demircigl. 2017, Ramsauer 2021)."

---

### Official Review · Reviewer_JBWE · 2025-06-30

**Clarity:** 2
**Significance:** 2
**Originality:** 2
**Rating:** 2
**Confidence:** 3

**Summary:**

The authors introduce a time-varying version of the Hopfield network, where a slowly varying temporal input can change the energy landscape and the resulting attractor the network converges to. The authors also claim *exponential memory* (but see below).

**Questions:**

1. I would ask the authors to rephrase the abstract and rest of the paper, to more clearly reflect the real bound of $\mathcal{O}(\min( \gamma^N, P))$. (Or explain to me that I am wrong if I am missing something).

2. Regarding Fig.5, is it possible / straightforward to adjust the model time scales to more closely match the observed data?

**Ethical Concerns:**

["NO or VERY MINOR ethics concerns only"]

**Final Justification:**

I appreciate the theoretical contributions to time-varying Hopfield networks. My main criticism was that the exponential memory part is very oversold in this paper. For one, the bound is only linear in the total amount of units in the network (which is somewhat hidden in the original manuscript), and second, similar bounds have been shown for very related networks. The authors have convinced me during the rebuttal that the terminology (exponential in feature units) is used commonly in the field. I would have considered raising my score, but statements by the authors such as the following make me hesitant to recommend accepting this paper without being able to have a second look at the manuscript:

>We want to clarify that the term "Exponential" in the title refers to the exponential interaction terms in the model

Which I simply find hard to believe considering the authors write in the manuscript statements like:

> To develop dynamic energy networks with exponential capacity, we incorporated a slow-changing
signal that interacts asymmetrically with an exponential capacity static energy network introduced in prior research

Are we really supposed to believe the authors talk about *exponential capacity* static energy network, but EDEN is an *exponential interaction* dynamic energy network?

If this paper gets accepted I do hope the authors take the suggestions by the reviewers very seriously and do make the fact that the bound is exponential only in the number of feature neurons very obvious.

**Limitations:**

yes

**Paper Formatting Concerns:**

no concerns

**Quality:**

2

**Strengths And Weaknesses:**

**Strengths**

The introduced model seems original to me, and the mathematical analyses using separation of time-scales for analysing the memory capacity appear well done. In general the paper reads well, and has nice visualisations (e.g., Figure 2a very nicely shows the changing energy landscape).

**Weaknesses**

I think that, unless I am missing something, the main claim of the paper title and abstract is not entirely reflected the content, as it the authors claim  *exponential memory capacity*, whereas the actual sequence memory capacity is $\mathcal{O}(\min( \gamma^N, P))$, where $P$ is the number of hidden units. In fact the model needs exponentially more hidden units for exponential memory (there is one hidden neuron for every memory to be stored). So arguably the model doesn’t really exhibit the exponential memory capacity that the authors claim. Note also that this bound on the number of memories is, as far as I can see taken from [13], where this additional linear dependence on the hidden units is indeed made very explicit.

The empirical demonstration is largely proof of concept (which is okay if the theoretical analysis are all sound).

---

> ### Author Rebuttal · Authors · 2025-07-31
>
> We thank the reviewer for positive comments on the mathematical analyses in our manuscript and the clarity of our figures. We address the weakness raised by the reviewer below.
>
> ### I think that, unless I am missing something, the main claim of the paper title and abstract is not entirely reflected the content, as it the authors claim _exponential memory capacity_, whereas...
>
> We thank the reviewer for the opportunity to clarify this point. The capacity argument is a common criticism of the literature of Dense Associative Memory models and is used in many works. Here, there is a subtlety in the definition of capacity - i.e capacity is defined as the maximum number of memories that can be stored as function of the dimensions in the state space (Line 197 in the paper) and not the actual neurons in the network. This is a subtle point and we have adjusted our discussion to make this clear. We have also introduced a limitation section to enumerate this weakness in the defining capacity in this manner.
>
> The reason why capacity is defined in this way is because the hidden neurons in Eqn (1) are often not explicitly written out as separate neurons, instead these variables are assumed to be part of dynamics of the visual neurons $(v_i)$ itself (by substituting $h_\mu$ in the dynamical equation of $v_i$). This requires an assumption that the synapses have many-body interactions (see \[1\] below). Even though this definition of capacity may seem to inflate the actual capacity, not all neural networks can accomodate arbitrary high hidden neurons for a fixed number of visual neurons, which is the main problem tackled by dense associative memory models. e.g. Classical hopfield networks can accomodate only a small fraction of the original visual neurons as hidden neurons.
>
> ### Regarding Fig.5, is it possible / straightforward to adjust the model time scales to more closely match the observed data?
>
> Thank you for the suggestion. It is not straightforward to match the parameters of our network with the biological data. The model we have and the energy-based networks in general are abstract models of memory storage. To our knowledge, there is currently no known setting of the parameters ($\alpha$'s and the timescales) that are validated from biological data. This is why we claim only that the biological connection is qualitative. If a procedure is devised to obtain these parameters from biological knowledge and use it for prediction, we may be able to predict some high level behavior of biological systems, this is a very interesting avenue to pursue in future work.
>
> \[1\] Krotov, Dmitry, and John J. Hopfield. "Large Associative Memory Problem in Neurobiology and Machine Learning." International Conference on Learning Representations. 2021

---

> > ### Comment · Reviewer_JBWE · 2025-08-01
> >
> > Taking into account the author's response and the other reviewers comments I am of the opinion that this paper needs a major rewrite before it can be accepted, so I will stick with my score.
> >
> > My suggestion would be to focus less on the "exponential" capacity (which as authors now confirmed is indeed actually linear in total number of units) and focus on the (strong!) contributions of the inclusions of temporal dynamics. I would personally even omit the word "exponential " from the title and architecture name.
> >
> > > The capacity argument is a common criticism of the literature of Dense Associative Memory models
> >
> > The Dense Associative Memory models paper [1] (from which the bound is taken), is not written in any way that overly promotes the claim of "exponential" memory, unlike your current submission. Additionally if your argument is in fact that previous papers used this bound, than one could also argue that it is not novel enough to be given such a spotlight in your paper.
> >
> > To end on a positive note, I do want to state that I do really like theoretical and empirical contributions of the time-varying dynamics in Hopfield networks, and do think that a paper focusing on that would provide a valuable contribution to the field.

---

> ### Author Response · Authors · 2025-08-02
> **Clarification and Addendum**
>
> We thank the reviewer for engaging with our rebuttal. We agree that the dynamic attractor dynamics is a central contribution of our paper. We agree that the term exponential for the capacity is misleading but, this is the terminology used in the modern Hopfield Network capacity. For further clarification, we urge the reviewer to consider the following updates we propose to the manuscript:
>
> 1. We mention in the abstract that the capacity. "The analysis of capacity for EDEN shows that it achieves exponential (in the dimensions of the feature space) sequence memory capacity ..."
> 2. We adjust the language in the introduction Line 79 to "The network capacity scales exponentially in the number of feature dimensions (the number of visible neurons)" and points where we write in the text "exponential in the number of neurons" to "exponential in the number of feature neurons".
> 3. We rewrite the bound as $P_{\max} = O(\min(\gamma^N, P))$ with $P$ as the number of hidden neurons in
> 4. We add a clarifying paragraph in the capacity section
> 		-"Note here that the exponential capacity as defined here takes into account only the number of neurons in the feature layer (the state space dimension) following similar capacity definition used in state associative memory models \[Krotov 2021, Demircigil 2017, Ramsauer 2021\]. The capacity is still linear in the number of hidden neurons, as there is one neuron per memory stored in the network. The hidden neurons are typically removed in the count as (1) they do not count meaningfully to the state space dimension - which contains $O(N)$ variables and can be substituted out in the dynamical equations if we were to allow for many body synapses \[Krotov 2021\], (2) Not all network interactions can accommodate exponential number of hidden neurons for a fixed number of feature neurons. Eg. the polynomial interaction sequence networks considered in \[Karuvally 2023, Chaudhry 2023\] can accommodate only $O(N^d)$ memories in the state space dimension, where $d$ is the polynomial interaction power. So, the scaling relationship of the number of memories with the number of visible neurons is an important consideration in the design of sequence networks."
> 5. We add a limitation section that includes the statement
> 	- "The exponential capacity we obtained requires an identical exponential increase in the number of hidden neurons, the question of how network capacity can be increased without taking hidden neurons into consideration is still a problem in both static (Hopfield-like energy) and sequential (dynamic energy) networks."
>
>
>
> ## The Dense Associative Memory models paper [1] (from which the bound is taken)
>
> We want to clarify that **we do not take the bound from any paper**. There have been similar use of exponential interactions in the dense associative memory models, we have tried to cover them in our introduction section. However these models are very different from the ones we study in our paper. We reworked these bounds and found them nearly identical in the form of scaling. The bounds we found, however, are not exactly identical as we had to re-derive the bounds under the new conditions of our model.
>
> ## I do want to state that I do really like theoretical and empirical contributions of the time-varying dynamics in Hopfield networks....
>
> We thank the reviewer for the positive comments and the potential impact of the contribution in the field.

---

> > ### Comment · Reviewer_JBWE · 2025-08-04
> >
> > I thank the authors for their clarification and after having gone through some additional references I now do agree that "exponential capacity" is used by related papers on modern Hopfield networks in a similar way as it is used in this contribution.
> >
> > I do still think the exponential point was somewhat oversold in the author's manuscript. E.g., assuming that it is conventional to define exponential memory like this, what are then the conventional models authors compare to in the abstract?
> > > The analysis of capacity for EDEN shows that it achieves exponential sequence memory capacity O(γN), outperforming the linear capacity O(N) of
> > conventional models.
> >
> > Nevertheless, I will take the authors clarification into account when updating my review for the final recommendation.

---

> > > ### Author Response · Authors · 2025-08-05
> > >
> > > We thank the reviewer for engaging with our response and for the point that related papers on modern Hopfield networks define capacity similar to how we do in our paper. We address the reviewer's question below:
> > >
> > > ### assuming that it is conventional to define exponential memory like this, what are then the conventional models authors compare to in the abstract?
> > >
> > > We compare against existing continuous-time memory models that tackles the sequential memory problem - Ref. 26, 34, 36 in the manuscript. These networks were previously found to have capacity lower than the dimensions of the feature space. We represent these networks using the reference network described in Equation 8 of the manuscript.
> > >
> > > [26] David Kleinfeld. Sequential state generation by model neural networks. Proceedings of the National Academy of Sciences of the United States of America.
> > > [34] Tomoki Kurikawa and Kunihiko Kaneko. Multiple-timescale neural networks: Generation of history-dependent sequences and inference through autonomous bifurcations.
> > > [36] Transitions among metastable states underlie context-dependent working memories in a multiple timescale network. In ICANN, 2021

---

### Official Review · Reviewer_q8MD · 2025-07-02

**Clarity:** 4
**Significance:** 3
**Originality:** 4
**Rating:** 5
**Confidence:** 5

**Summary:**

The paper proposes a method to let a Hopfield Network evolve temporally via energy functions of short timescales. Such a formulation is derived by considering two distinct sets of entities that govern the evolution of the energy of the Hopfield system, i.e. fast population neurons (borrowed from the vanilla Hopfield model) and a set of slow population neurons, that attempt to remember previous memory state(s) even after the energy has stabilized over the current state. Accordingly, properties of the system such as energy dynamics (and consequent transitions between memory states) and escape time are analyzed both theoretically and empirically over a simple MNIST dataset. Strong agreement between the theoretical results and empirical findings is noted. Theoretical results on the capacity are also derived. Finally, the paper also discusses a close resemblance of the proposed energy dynamics with biological episodic memory systems.

**Questions:**

1. What is $z_\mu$ in Equation 12 (Appendix)? How does Equation (15) follow from Equation (14)?

2. In line 174, why are only energy corresponding to two consecutive memory states considered to compute the exact time? Technically, shouldn't this be summed over all prior states? Have only two consecutive memory states been considered under the assumption (if this isn't an assumption, please show why it holds in general) that contributions from states before the penultimate one are negligible?

3. To derive capacity bounds, I am assuming that the patterns are assumed to be binary (for otherwise Rademacher variables would not be suitable to capture the representation of the patterns). Am I right in assuming this?

4. What is $\hat{v}$ in Equation 48?

5. Just to confirm, in line 441, $\alpha$ is indeed $\alpha_s$, right?

6. With respect to Equation (59), I am guessing $\beta_{\alpha r}\beta_{\alpha} = \beta_{\alpha_s}\beta_{\alpha_c}$, based on above. Can the authors add a short comment stating when this term is less than 1, based on their definition of $\beta_x$?

7. In Equation (23), what is $v_i$ chosen as that enables tractable integration? This may not be true for generic $v_i$, right?

8. In Equation (41), if $\lambda$ is being derived by equating the coefficients in Equation (40), Equation (41) is not correct. How is it being derived?

**Ethical Concerns:**

["NO or VERY MINOR ethics concerns only"]

**Final Justification:**

This is a theoretically strong, and empirically verified paper in the general field of enery based memory models. The authors did a great job clarifying all my initial concerns. I hope to see this paper accepted at this conference.

**Limitations:**

I don't see a separate limitations section in the paper. The authors are advised to do this.

**Paper Formatting Concerns:**

None.

**Quality:**

4

**Strengths And Weaknesses:**

Strengths:
1. The paper has been presented very well, in terms of readability, appropriate figures, plots and ordering of sections.
2. The paper proposes a novel Hopfield Energy network formulation for temporal domains, by evolving and piecemealing the energy function over multiple times scales. The paper analyzes a few salient properties of the system in detail, such as its energy dynamics, and escape time.
3. The above approach is strongly grounded in a number of theoretical results, and is built from first principles of energy dynamics.
4. The authors also do a good job of justifying each of the components of the proposed formulation with biological counterparts.


Weaknesses:
My main concerns are with the first two points under the Related Work and first point of the Experiments section below. These are followed by a number of technical questions in the Questions section, followed by other concerns mentioned here. I am eager to increase my score, especially if the first two concerns can be addressed. I thus suggest the authors to prioritize addressing the concerns according to the order I have indicated.

Related Work:
1. Line 88 of the main paper suggests that the 'Long Sequence Hopfield Memory' paper does not use energy arguments. While it is unclear what the authors mean by this, it seems to suggest as though they haven't used energy functions rigorously enough, which I think is untrue. They also considering timing delays (albeit not as thoroughly as in this paper) and asymmetric networks. So rather than artificially mellowing it down too much, I suggest the authors spend just a few sentences more concretely describing how their approach differs from the 'Long Sequence Hopfield Memory' paper.

2. This work seems to exhibit a lot of similarities to https://proceedings.mlr.press/v202/karuvally23a.html -- please consider explicitly justifying how your work differs from theirs.

3. I also suggest suggest adding https://journals.plos.org/ploscompbiol/article?id=10.1371/journal.pcbi.1011183, https://proceedings.neurips.cc/paper_files/paper/2023/hash/8a8b9c7f979e8819a7986b3ef825c08a-Abstract-Conference.html as a citation and briefly justifying how your work differs from theirs in its core contributions.

Experiments:
1. In  my understanding, it appears that the method has been tested only on one dataset, i.e. MNIST. While I understand that this is perhaps largely a theoretical work, I strongly suggest showing your methodology and showing results like Figure 2 for at least 2-3 other datasets. You can use datasets from standard papers such as 'Hopfield Networks is All You Need', 'Associative Memories via Predictive Coding," or any other such papers.

2. Line 142 of the paper claimed that no non-convergent behavior was observed, but I am led to believe that this was because MNIST is a very simple dataset. Even the 'Hopfield Networks is All You Need' paper suggested that Hopfield Nets could hit unstable local minima or averages of local minima, and this can be observed via some experiments also. In light of this, please either show via datasets chosen above that non-convergence practically never happens regardless of dataset or show some cases where this failure does happen.

Technical:
1. In line 443 (Appendix), just after the statement about modelling via Rademacher r.v's, it appears that a minus sign has crept into the exponent of the term in the denominator which wasn't present earlier. This seems to affect the bound critically as well. Please clarify or fix this.

2. The first line of page 15 (Appendix) requires justification. I am not sure how the assumption of modelling via Normal r.v's can be directly made (there is no reference provided for this either).

3. The way Equation (24) follows from Equation (23) does not seem straightforward to me. Please explain more clearly about how Equation (24) is arrived at.

Minor:
1. Please change "Appendix 24" to "Equation 24 in the Appendix" or some equivalent of that in line 203 of the main paper.
2. In line 165 of the main paper, please change $\mu$ in the superscript of $\zeta$ with $\nu$.

---

> ### Author Rebuttal · Authors · 2025-07-31
>
> The thank the reviewer for the comments and highlighting the grounding of our core contributions in rigorous theory. We also thank the reviewer for positive comments on the presentation. We appreciate the level of detail and meticulousness involved in formulating this review and the attention to the math that enabled finding typos in the document. Prior to addressing the concerns, we want to point out a key assumption of Markovian transitions we make in the theory that enables us to obtain analytically tractable results.
> # Markovian Assumption
>
> In line 195, we make the assumption that the capacity is computed at the phase change boundary, but the key assumption required to obtain the analytic form of Eqn (10) for the capacity is that the state transitions encoded in the network are Markovian and the network stays in a memory state long enough for the previous memory state to dissipate. i.e. Eqn 43 in the appendix is such that $s_i(t) = \sqrt{r} \xi^{(2)}_i + \xi^{(1)}_i (1 - \sqrt{r} ) \approx\sqrt{r} \xi^{(2)}_i$, which happens when $r=\alpha_s/\alpha_c$ nears 1, but does not necessarily have to be exactly 1, where the escape time is infinite. Due to exponential decay, the information is lost in an exponential rate, so the approximation error will be exponentially bounded.
>
> In sequential networks where we observe *stable* state transitions, this condition seems to be satisfied. Without this condition, the network will easily recall spurious memories .i.e the network may converge to a spurious memory that is a combination of $\xi^{(2)}$ and $\xi^{(3)}$ when transitioning from $\xi^{(2)}$ because some signal of $\xi^{(1)}$ is left in the state. We have amended our discussion to clarify this assumption better. Most of the questions posed are related to this.
>
> ## Line 88 of the main paper suggests that the 'Long Sequence Hopfield Memory' paper does not use energy arguments.
>
> Thank you for raising this concern. There are no energy arguments in the "Long Sequence Hopfield Memory" paper. i.e there is no energy function defined for their network, the starting point of the paper is an asymmetrically connected Hopfield network. This is because in their case, the authors consider a discrete sequence network where there is no separation of timescales to define an energy function. We have amended our discussion to include the discrete-continuous difference.
>
> ## this work seems to exhibit a lot of similarities to...
>
> We thank the reviewer for pointing out the work. The network and the analysis we performed go beyond the aforementioned work in two ways - (1) compared to the above paper, we provide "quantitative" and analytic guarantees on the properties of the network. (2) The networks considered in the Karuvally paper are polynomial activation networks and did not provide mathematical guarantees of capacity, while we provide both empirical and theoretical guarantees of network scalability.
>
> ## I also suggest suggest adding...
>
> We thank the reviewer for the suggestions. We have amended our introduction to include comparisons to these two paper. We identify two differences compared to the referenced works. In the above two papers, the authors consider an abstract network that learns to encode temporal stimuli, where the plasticity rules play a role in encoding temporal information. In our setting, we analyze network dynamics where the temporal information is hardcoded. There are no plasticity rules in our models, but this is an interesting future avenue to pursue. The second difference is that the model we consider is a continuous time model, which enables applying the energy formulation to understanding the network. Discrete models lack this energy formulation for dynamical behavior. The continuous-discrete difference may also be one of the reasons we find qualitative behavior similar to biological networks.
>
> ## In my understanding, it appears that the method has been tested ...
>
> Thank you for the suggestion, we will add more datasets in the final paper submission in the appendix. As rightly pointed out, the aim of the paper was to obtain analytically tractable features of the network.
>
> ## Line 142 of the paper claimed that no non-convergent...
>
> Thank you for pointing this out. It is correct that in the regimes we tested, there were no spurious states. However, if the Markovian assumption is violated and the network is forced to transition before the memory of the previous state is lost, then the low energy state becomes a combination of two encoded memories. e.g. for the sequence of transitions $\xi^{(1)} \to \xi^{(2)} \to \xi^{(3)}$, during the transition from 2 to 3, if the slow signal still has a trace of 1, then a spurious minima is formed as a combination of $\xi^{(2)}$ and $\xi^{(3)}$. Excluding this case is a key assumption we make in the theory when the capacity is analyzed close to the phase change boundary. These spurious states become relevant when the ratio $\alpha_s/\alpha_c$ is far from the phase boundary.
>
> ## In line 443 (Appendix), ....
>
> We thank the reviewer for pointing this out. The minus sign is actually an $\alpha$ that should be corrected in all the denominators of line 443 starting from $Z = \sum_{\nu \neq 3} \frac{\exp(\alpha( r \langle \xi^{(\mu)}, \xi^{(3)} \rangle + \langle \xi^{(\mu-1)}, \xi^{(2)} \rangle ))}{\exp(\alpha(1+r)(N-1))}$. We apologize for the oversight and is corrected in the manuscript. The bound is still correct.
>
> ## The first line of page 15 (Appendix)...
>
> Thank you for pointing this out, we have amended the discussion to clarify this point now. Since $\xi^{(\mu)}_i$ is Rademacher distributed, $\xi^{(\mu)}_i \xi^{(\nu)}_j, \mu \neq \nu$ is also Rademacher distributed. Next, the sum of Rademacher random variables are binomial distributed, which when $n$ is large, can be approximated by a Normal distribution. These follow from the standard properties of the distributions.
>
> ## The way Equation (24) follows from Equation (23) does not seem straightforward...
>
> For equation 24, we assume that the network transitions from 1->2 almost instantly, so in order to obtain an analytically tractable form for this, we set $t_0=0$, $s_i(t_0)=\lambda \xi_i^{(1)}$ and $v_i(s)=\xi_i^{(2)}$, a constant, for the duration.  This approximation is valid as long as two conditions are satisfied (1) $\mathcal{T}_d$ is larger than $\mathcal{T}_f$, that is, there is a separation of timescales and the transient transition between the two memory states are negligible and (2) Markovian state transition, that is the network stays in $\xi_i^{(1)}$ long enough till $\xi^{(P)}_i$ is fully lost from the state. The lambda is assumed, and then later found using the energy function dynamics.
>
> ## Question 1: What is $z_\mu$ in Equation 12 (Appendix)? How does Equation (15) follow from Equation (14)?
>
> $z_\mu = \exp(h_\mu)$ is a short hand for the exp terms. I have defined it now in the manuscript. We apologize for the oversight. We appreciate the reviewer pointing out the typo in Equation 14. The summation over $i$ is is not ordered correctly. Eqn 14 is $\frac{dE}{dt} = \sum_{i} \frac{d v_i}{d t} ( v_i  - \sum_{\mu} \frac{z_\mu}{\sum_{\nu} z_\nu} \xi^{(\mu)}_{i} ) + ...$ and $\frac{d E}{dt} = - \sum _{i} \mathcal{T}_f \left(\frac{d v_i}{d t}\right)^2 + ...$.
> There is no change in the discussion or the results because of these typos.
>
> ## Question 2: In line 174, why are only ....
>
> This is a result of the Markovian assumption. If the network operates farther from the phase change boundary, the influence of prior states become non-negligible and the network starts showing spurious memories.
> ## Question 3: To derive capacity bounds ...
>
> Yes, the capacity bounds are derived for Rademacher distributed patterns.
>
> ## Question 4: What is $\hat{v}$  in Equation 48?
>
> Thank you for pointing out this typo - $\hat{v}$ is the slow variable $s_i$.
>
> ## Question 5: Just to confirm, in line 441, $\alpha$  is indeed $\alpha_s$, right?
>
> Yes, that is correct. The subscript is removed only for exposition.
>
> ## Question 6: With respect to Equation (59), ...?
>
> Yes, this is added now. $\beta_x = \frac{\cosh(x)}{\exp(x)} < 1$ when $x>0$.
>
> ## Question 7: In Equation (23), ....
> $v_i(s)=\xi_i^{(2)}$ a constant, for the duration. The assumption here is that the the slow timescale enables ignoring the transient when transitioning from $\xi_i^{(1)} \to \xi_i^{(2)}$.
>
> ## Question 8: In Equation (41),...
>
> To derive the expression for $\lambda$ from Eqn 40, we make use of our Markovian assumption. i.e the memory of the old memory state is lost by the time the network reaches the escape time. From there, we take the coefficient of $\xi_i^{(2)}$ and equate it to $\lambda$ which is defined in Line 406 as a factor that controls to what extend the previous state is reflected in the slow variable before state transition occurs. This assumption becomes valid closer to the phase change bounday $\alpha_s/\alpha_c$ approaches 1, but not exactly 1.
>
> # Minor Typos
>
> We thank the reviewer for finding these typos. These are corrected in the manuscript.

---

> > ### Comment · Reviewer_q8MD · 2025-08-05
> > **Requires a Little More Fixing**
> >
> > I thank the reviewers for their sincere attempt in responding to all my comments. I also apologize for my delayed response back to the authors. Most of the responses seem satisfactory. In general, as the authors probably themselves recognize by now, it is quite important to mention the Markovian assumption here for most of the theoretical connections to hold, so this should be suitably inserted into the main text of the paper preferably; if not in the main text, then at least in the Appendix for sure. I also suggest the authors to correct all the major and minor typos that they have discovered now, into the final manuscript. Apart from this, I will only post the points that I feel still require further work/clarification (in decreasing order of importance):
> >
> > ### In my understanding, it appears that the method has been tested ...
> > Please specify which exact datasets you plan to include. I would be much more comfortable going ahead with this discussion if you could actually test it on some dataset and provide the results for it. Ideally this should have been done by now, but please make sure it is at least done by the end of this Discussion phase (and the results are shown and approved by the Reviewers here).
> >
> > ### Question 8
> >
> > If that is the case then shouldn't the coefficient of $\xi_i^{(1)}$ be equated to $\lambda$ instead of that of $\xi_i^{(2)}$? Since $\xi_i^{(1)}$ is the "old" pattern here.
> >
> > ### this work seems to exhibit a lot of similarities to...
> >
> > Thank you for clarifying this. Please mention this in the paper nonetheless.
> >
> > ### Line 88 of the main paper suggests that the 'Long Sequence Hopfield Memory' paper does not use energy arguments.
> >  Please mention what you explained in the rebuttal here, of their not using an explicit energy function as defined in their paper, instead of using the vague term 'energy arguments.' It is hard to believe that the update rule derived would not have any grounding in any sort of energy function, even though it is not explicitly defined in the paper.
> >
> > ### The first line of page 15 (Appendix)...
> > Sure, but please consider adding a reference for this (even a standard textbook that mentions these properties should do).
> >
> >
> > I look forward to increasing my score if these concerns are properly addressed.

---

> > > ### Author Response · Authors · 2025-08-06
> > >
> > > We thank the reviewer for engaging with our responses. All the changes we proposed in our rebuttal responses is already made in our revised version. The Markovian assumption is clarified in the capacity discussions of the main text and a section is included in the appendix for details of the assumption. We have also added the comparison to the Karuvally 2023 paper in the main text.
> > >
> > > **Please specify which exact datasets you plan to include...**
> > >
> > > We have added experiments that show sequence retrieval in movies from two datasets (1) moving MNIST (64x64 size images), which was previously used in Long Sequence Hopfield Memory paper and (2) the KTH action dataset (subsampled to 100x100 resolution). We have movies showing the evolution of the feature neurons in the network when the movie frames are encoded as sequences which we are happy to share. Unfortunately, NeurIPS policy does not allow links to images/movies during the review responses. We plan to add these movies in the supplementary materials. We believe that the core contributions to the theory of sequence networks is still relevant even in the absence of these videos.
> > >
> > > ## If that is the case then shouldn't the coefficient of $\xi^{(1)}_i$ be equated to $\lambda$ instead of that of $\xi_i^{(2)}$ ? Since  is the "old" pattern here.
> > >
> > > This is a good question. I think the cause for the confusion is the grammar we used in defining $\lambda$ in line 406. We propose the following update to the language in the text "For a transition $\xi^{(1)} \to \xi^{(2)}$, $\lambda$ is a factor quantifying the extend to which the previous state $\xi^{(1)}$ is present in the slow population when the state transition occurs. Using the Markovian assumption, we assume that $\xi^{({P})}$ is negligible in the slow population when the transition occurred".
> > >
> > > We start our analysis just after the transition $\xi^{(1)} \to \xi^{(2)}$ happened where the "old" pattern is indeed $\xi^{(1)}$. The escape time we compute is for the state transition $\xi^{(2)} \to \xi^{(3)}$. Now, the definition of $\lambda$ is used again as before but on the transition $\xi^{(2)} \to \xi^{(3)}$ similar to above, except now the "old" pattern is $\xi^{(2)}$. So the substitution is correct.
> > >
> > > An alternate way to think about $\lambda$ is by imagining a factor corresponding to a memory in the slow variable that increases as the network stays in a meta-stable memory state. Now, this factor *ideally* would reach 1 asymptotically over time, while any "old information" exponentially decays to 0 at which point the fast variable escapes the memory state. Instead of exactly $1$, we use a factor $\lambda$ and compute what this is based on the parameters we have in our model. A sanity check is to verify if the factor at escape time in the most ideal Markovian case is very close to 1, which we indeed find in our analysis.
> > >
> > > ## It is hard to believe that the update rule derived would not have any grounding in any sort of energy function, even though it is not explicitly defined in the paper.
> > >
> > > We have clarified this in the discussion of Long Sequence Hopfield Network in the revised manuscript. The "energy argument" is that the network evolves such that over time, there is some scalar function of its state that always decreases. This special scalar function is the potential/energy/Lyapunov function for the dynamics of the network. In the discrete system that was considered in the Long Sequence Hopfield Memory paper, an energy function does not exist. In EDEN, we have energy functions in the short timescales only because of the timescale separations of the asymmetric interaction terms.
> > >
> > > ## The first line of page 15 (Appendix)
> > >
> > > We have added the reference "An Introduction to Probability Theory and Its Applications by William Feller Vol I". There is a discussion of the large sum behavior of binomial variables. We have also added a Lemma to the Appendix for how large sums of independent Rademacher variables can be approximately Gaussian.

---

> > > > ### Comment · Reviewer_q8MD · 2025-08-07
> > > > **Satisfied**
> > > >
> > > > I thank the authors for healthily engaging in discussions with me and clearing all my concerns. With respect to the last response of the authors to my questions, I ask them to update the explanation about $\lambda$ as they have explained it here, in the manuscript. I am raising my score to a 5. I really hope to see this paper accepted at NeurIPS'25. Good luck!

---

### Official Review · Reviewer_rFqE · 2025-07-03

**Clarity:** 3
**Significance:** 2
**Originality:** 3
**Rating:** 4
**Confidence:** 4

**Summary:**

The paper proposes Exponential Dynamic Energy Networks (EDEN), a two-population Hopfield variant in which a slow modulatory population continually reshapes the energy landscape seen by a fast, high-capacity Hopfield core. This drift causes the network to leave one stored attractor and march through a pre-wired cycle of memories. The authors (i) establish a Lyapunov energy for the fast dynamics, (ii) derive a closed-form mean escape-time that marks a phase boundary between “static” and “dynamic” regimes, and (iii) argue that the model can store an exponentially long sequence of memories via a mean-field calculation. Synthetic experiments fit the escape-time law (for binary Rademacher patterns with N ≤ 35) and the authors draw analogies to hippocampal time/ramping cells.

**Questions:**

1. Validity and Scale of the Capacity Claim: This is my primary concern. Can you provide evidence that exponential capacity holds in the model's actual operating regime? This could be done in two ways: (a) empirically demonstrate the scaling behavior for α_s /α_c < 1 (e.g., 0.9) at a much larger scale, such as N ≥ 256; or (b) provide a theoretical argument that bounds the error of the mean-field approximation and shows that the exponential nature is preserved away from the phase boundary. Convincingly addressing this is critical. If this is not possible for some reason, please explain it to me. I apologize if I missed something basic.

2. Hidden Layer Scaling: Storing γ^N memories naively implies an exponential-size hidden population. Do you instantiate it explicitly? Please spell out memory/time complexity and any tricks that keep things tractable. This is a common criticism for the biologically plausible implementations of the Exponential Dense Associative Memory.

3. Storing Multiple Sequences: The model in its current form seems to store only a single, pre-wired circular sequence. How would the architecture need to be modified to store multiple distinct, non-interfering sequences? How does EDEN cope with real-valued or noisy or correlated patterns (e.g., MNIST with additive noise)?

4. The proposed link to neuroscience is intriguing but qualitative. The model predicts a sharp relationship between the timescale ratio T_d / T_f and the system's dynamics. Could you attempt to fit this relationship to actual neurophysiological data, such as the ramping cell activity from Umbach et al. (2020) or a similar dataset? A quantitative match, even a rough one, would make the biological relevance claim substantially stronger. I understand this is likely difficult in the short rebuttal period, but if you can offer any further backing to the biological plausibility I believe this can improve the strength of the paper.

A convincing response to Question 1 is essential. Addressing that question along with strong answers to any two of the others (2, 3, or 4) would be sufficient for me to raise my score to a borderline or clear accept.

**Ethical Concerns:**

["NO or VERY MINOR ethics concerns only"]

**Final Justification:**

They addressed the questions and comments and I see value in their proofs. I believe other reviewers also noted that their citation of previous literature was somewhat sparse and that the idea of storing exponentially long sequences had already been achieved prior in Chaudhry et al or having a dynamic energy function with polynomially long sequences in Karuvalley et al. I think the mention of biological plausibility when discussing exponentially many hidden neurons doesn't make too much sense, but that is a problem of the field rather than this paper alone.

If the authors properly situate their paper in the broader literature context in the final version, it would serve to highlight the paper's primary strength which is its theoretical arguments.

**Limitations:**

Yes, but the discussion is insufficient. The most critical limitations (the simplified setting for the capacity proof, the lack of a learning rule, the single-sequence architecture, and the scalability concerns) are not consolidated or discussed in the main text. A dedicated "Limitations" section is needed to give readers an honest and clear assessment of the work's current scope and boundaries.

**Paper Formatting Concerns:**

I didn't see anything egregious.

**Quality:**

2

**Strengths And Weaknesses:**

The paper's core theoretical contribution is sound. The extension of the energy paradigm to the temporal domain via a slowly drifting landscape is a clean, powerful concept. The derivation of the escape time in Equation 5 is the key highlight for me, providing a clear and principled link between the model's parameters and its dynamic behavior which lines up well with the toy experiments.

However, my enthusiasm for the theory is significantly dampened by the validation of the paper's headline claim: exponential sequence capacity. The claim rests on a fragile foundation for two distinct reasons. First (as noted by the authors), the analytical proof is performed only at the phase transition boundary (α_s / α_c → 1), a special case where escape times are infinite and the model ceases to function as a sequential memory system. It is not at all clear that this result holds in the dynamic regime. Second, the experimental validation leans entirely on a Gaussian mean-field approximation that is only tested on tiny networks (N ≤ 35). Without either a larger-scale experiment or a theoretical bound on the approximation error, the capacity result feels more aspirational than proven.

Furthermore, the work is not well-situated within the broader context. There is no task-level comparison to modern sequential models (RNNs, Transformers) or even recent sequence-Hopfield baselines (e.g., Chaudhry et al., 2023), making it difficult to assess the practical significance. While the use of a drifting energy landscape is a neat twist, the originality is tempered by the fact that asymmetric, multi-timescale Hopfield models have existed for decades (e.g., Kleinfeld ’86; Sompolinsky & Kanter ’86), and a more explicit discussion of what is new versus what is borrowed would strengthen the paper's positioning. Furthermore, the authors should also read and consider citing highly relevant recent work on analysis of dynamic energy landscapes for sequential Hopfield networks (Karuvally, Sejnowski,  Siegelmann 2023).

With regard to the clarity and writing of the paper, the narrative flows well, the figures are helpful, and the appendix is thorough. However, key assumptions—perfect weight learning, binary memories, strict time-scale separation—are scattered in footnotes and supplements. A concise “Limitations” box in the main text would save readers a scavenger hunt.

---

> ### Author Rebuttal · Authors · 2025-07-31
>
> We thank the reviewer for highlighting that our proposed extension of the **energy paradigm to the temporal domain via a slowly drifting landscape is a clean, powerful concept.** and positive comments about our derivation of escape times. We first clarify our key assumption when deriving the capacity formula and address the weaknesses below.
>
> ### Clarification of Markovian State Transition Assumption
>
> In line 195, we make the assumption that the capacity is computed at the phase change boundary, but the key assumption required to obtain the analytic form of Eqn (10) for the capacity is that the state transitions encoded in the network are Markovian and the network stays in a memory state long enough for the previous memory state to dissipate. i.e. Eqn 43 in the appendix is such that $s_i(t)\approx\sqrt{r} \xi^{(2)}_i$, which happens when $r=\alpha_s/\alpha_c$ nears 1, but does not necessarily have to be exactly 1, where the escape time is infinite. Note that Eqn (10) for the capacity has $r$ as a parameter so the equation can be used to probe what happens close to the boundary (We chose 0.999 for our simulation). In this limit $(1-\sqrt{r}) = 5 \times 10^{-4}$ which is assumed as 0.
>
> In sequential networks where we observe stable state transitions, this condition needs to be satisfied. Without this condition, the network will easily recall spurious memories .i.e the network may converge to a spurious memory that is a combination of $\xi^{(2)}$ and $\xi^{(3)}$ when transitioning from $\xi^{(2)}$ because some signal of $\xi^{(1)}$ is left in the state. We have amended our discussion to clarify this better.
>
> ### Limitations
>
> We add a limitations sections to collect the assumptions we make for the formulated theory in our paper.
>
> As a theory of dynamical behavior of a non-linear system, we make key assumptions that simplify our mathematical analysis. (1) The synaptic strengths of the neuron interactions are fixed and does not vary during training, in actual biological systems synaptic strengths can change due to short and long term potentiation effects and consolidation (2) The timescales of symmetric and asymmetric interactions are separate - this allows use to treat the asymmetric part as slowly evolving and change the energy function of the symmetric network is response. In human brains, there are different timescales for information processing but the timescales may not be perfectly separated as a distinct slow population of asymmetric connections and fast population of symmetric population, (3) Binary memory - we assume Rademacher distributed patterns for theoretical exposition following related works in the field, although the theory can be similarly worked out for other distributions (4) Markovian State Transition - in deriving our capacity bounds, we assumed that the network spends enough time in a memory state that the historical trajectory information is lost and the state transitions are purely Markovian in nature. Further, the capacity bounds we formulated shows how the maximum number of memories (number of hidden neurons) scales with the number of visual neurons ($v$) following previous results in the field. The number of hidden neurons required for storage however grows only linearly in the number of hidden neurons.
>
> ### Limitation 1: the analytical proof is performed only at the phase transition boundary (α_s / α_c → 1), a special case where escape times are infinite and the model ceases to function as a sequential memory system.  It is not at all clear that this result holds in the dynamic regime.
>
> See our clarification above. The key assumption we use to compute the capacity is that Eqn (43) in the appendix for the slow signal can be written as $s_i(t) = \sqrt{r} \xi^{(2)}_i + \xi^{(1)}_i (1 - \sqrt{r}) \approx \xi^{(2)}_i$ which happens when $\sqrt{r}$ nears $1$ that is **close to the boundary** and not exactly on the boundary. The factor $r$ still occurs in our capacity equation and can be used to probe what happens slightly away from the boundary.
>
> For example, we do not assume anything about $\mathcal{T}_d/\mathcal{T}_f$, so these can be tuned to have the capacity guarantee in a reasonable timeframe. We have amended our discussion to make this nuance clear. In our simulations, we used $r=0.999$ not exactly $1$ and the $\mathcal{T}_d$ was set at $20$, which produced sequential transitions in a reasonable time in the network **which is the dynamic regime**. For the dynamic regime, the ratio just needs to be less than 1.
>
> ### Second, the experimental validation leans entirely on a Gaussian mean-field approximation that is only tested on tiny networks (N ≤ 35). Without either a larger-scale experiment or a theoretical bound on the approximation error, the capacity result feels more aspirational than proven.
>
> The theory is derived in the limit of large N (asymptotic regime) so the bound becomes more valid as N is increased, The reason we constrain to N<= 35 is because the capacity for these networks are under the 1M (Million) memory limit. The 1M  constraint is the limitation of the simulation hardware available to us. Our network still works for large scale tasks - for example, we simulated a 784 (28x28) unit network to obtain Fig 2 in the paper that uses MNIST digits. The capacity for the 784 network is way higher than 1M memories, MNIST has only 60k samples.
>
> ### Furthermore, the work is not well-situated within the broader context. There is no task-level comparison to modern sequential models (RNNs, Transformers) or even recent sequence-Hopfield baselines (e.g., Chaudhry et al., 2023), making it difficult to assess the practical significance.
>
> The intention of the paper was to describe salient features of an Exponential interacting network using a novel dynamic energy function. Our contributions are primarily in understanding these network mathematically in an analytically tractable way and not to establish any superiority over existing models in practical tasks. This is still an important avenue to pursue, which will be done in future works. We urge the reviewer to consider the merits of the paper within the context of our contributions.
>
> ### While the use of a drifting energy landscape is a neat twist, the originality is tempered by the fact that asymmetric, multi-timescale Hopfield models have existed for decades (e.g., Kleinfeld ’86; Sompolinsky & Kanter ’86), and a more explicit discussion of what is new versus what is borrowed would strengthen the paper's positioning.
>
> As rightly pointed out, asymmetric multi-timescale Hopfield networks have existed for decades, however, to our knowledge computation of the salient features of the network were analytically intractable, and construction of networks with exponential capacity provably has not been conducted. We thank the reviewer for suggesting the citation of Karuvally et. al. (we have added it now with discussion in the introduction). There are two key changes to the above paper we made. (1) While the above paper provided qualitative descriptions of dynamic behavior using the energy approach, we use quantitative approaches that is analytically tractable, (2) The models proposed in the above paper were polynomially interacting and hence had power law capacity with no proofs of the capacity claims, here we make the interactions exponential, and prove that the capacity is exponential in the number of feature neurons.
>
> ### Questions
>
> ### Question 1: Validity and Scale of the Capacity Claim
>
> See the above discussion in the Weakness section. To further support the assumption, note that the slow neuron dynamics (in Eqn 26) shows that previous memory state decays exponentially over time. We further analyzed the capacity equation in the presence of one previous memory (non Markovian with history 1), in which case a spurious pattern can be shown to form if the escape time is not long enough. So, setting the operating point close to the boundary is necessary for the stable retrieval of sequential patterns.
>
> ### Question 2: Hidden Layer Scaling
>
> We thank the reviewer for pointing this out. The hidden population size is exponential, which is the current practice in Exponential Dense Associative Memory models. This is one of the reasons why the capacity experiments are constrained to networks <= 35 and upper bounded at 1M memories. To find the capacity, we use a binary search style algorithm where a range of patterns from 1 to 1M is searched to find the upper bound. With this change, the capacity experiment time complexity is only logarithmic in the number of memories which is favourable for such dense networks. Beyond this, there are no additional tricks, so the memory also scales linearly in the number of memories.
>
> ### Question 3: Storing Multiple Sequences
>
> The key assumption needed is Markovian sequences, storing multiple sequences does not need a significant change. circular sequence is only used for convenience in the exposition. We refer the reviewer to Karuvally et. al. where more general connectivity patterns is considered.
>
> ### Question 4: Link to Neuroscience
>
> The focus of the current paper was on understanding salient features of multi-timescale non-linear dynamical system analytically. As of now, the biological connection is only qualitative. The theory and model we developed is in a higher abstraction level compared to biophysical models like Hodgkin-Huxley models. Due to the abstraction, the high level qualitative features may become evident from the model, but not the fine quantitative details needed to fit the parameters. This is a common weakness of models neural models higher in abstraction.

---

### Official Review · Reviewer_kb6a · 2025-07-08

**Clarity:** 3
**Significance:** 3
**Originality:** 3
**Rating:** 5
**Confidence:** 3

**Summary:**

This work proposes a new kind of dense associative memory (DAM) model for storing and retrieving sequent memory. The proposed model contains two kinds of neurons operating at different time-scales: fast neurons and slow neurons. The slow neurons are organized in a classical dense associative memory network, while the slow neurons modulate the energy landscape the fast neurons are trying to minimize. Thus, the whole system can be seen as a time-evolving DAM. The authors show the following:

- The system spends a number of time steps trapped in a specific memory, and then escapes into another memory, etc. The separation between the time steps (the so-called escape time) is analytically calculated in terms of macrospic parameters that appear in the equations describing the evolution.

- A precise phase-transition is computed as a function of some of the the aforementioned macroscopic parameters, more precisely, in terms of the ratio $r:=\alpha_s / \alpha_c$ of energy scales in the model. For $r>1$ the system only contains a fast neurons, and is stuck in the classical static regime. For $r<1$, the system contains a transition into a dynamical regime.

- It is shown that the model has overall exponential memory capacity.

- Finally, a biological connection is made between the proposed model and human episodic memory.

**Questions:**

See the questions outlined in the "Weaknesses" above.

**Ethical Concerns:**

["NO or VERY MINOR ethics concerns only"]

**Limitations:**

yes

**Paper Formatting Concerns:**

Nothing to declare here

**Quality:**

3

**Strengths And Weaknesses:**

### Strengths

Though this is not the first time DAM's have beeng proposed for handling sequential memory, this work clearly expands the current scope of DAM's by introducing a time-variable. The resulting model neatly captures the concept of sequential memory. Rigorous analysis are conducted to compute key quantities like escape times (between one memory state and the next). Finally, from the biological plausibility angle, strong links to human episodic memory are made.


### Limitations
- ICLR is an ML conference (not a biology conference). What does this model have to say about ML? Does it explain any ML phenomenon or can it be a basis for a new time of ML model? Any experimental results in this direction?
- What is the motication of assuming that the sequential memories are organized in a circle? How robust are the results to this somewhat adhoc (but technically convenient?) construction?

---

> ### Author Rebuttal · Authors · 2025-07-31
>
> We thank the reviewer for the positive comments on the manuscript and suggesting that our model **clearly expands the current scope of DAM's by introducing a time-variable**, the theory we developed to compute quantities like escape time is **rigorous**, and the connection to **biological features is strong**. Below we address the concerns raised by the reviewer.
>
> ### Limitation 1:  - ICLR is an ML conference (not a biology conference). What does this model have to say about ML? Does it explain any ML phenomenon or can it be a basis for a new time of ML model? Any experimental results in this direction?
>
> It is our understanding that NeurIPS is an interdisciplinary conference where researchers from ML and neuroscience gather to share ideas. While we have not discussed the merits of our model in the ML domain, identical works in static Dense Associative Memory models have shown promise in both improving and understanding relevant ML ideas. Eg. DAM has strong connections to transformers and can be used as external memory stores. Additionally, recent works have generalization performance of diffusion models can be understood via phase transitions observed in energy function approach.
>
> ### Limitation 2 - - What is the motication of assuming that the sequential memories are organized in a circle? How robust are the results to this somewhat adhoc (but technically convenient?) construction?
>
> The primary motivation for the assumption is to simplify the theoretical exposition. The main assumption we use is that the sequential transitions are Markovian. i.e. the memory to transition to determined purely from the current memory and not the history of memory transitions in the network. eg. when network moves from 1->2->3, the transition 2->3 is independent of whether the network started from 1 then transitioned to 2 or the network initially started from 2. As such, all theoretical results should hold as long as the escape time is long enough for the network to forget this history.

---

> > ### Comment · Reviewer_kb6a · 2025-08-04
> >
> > Thanks for the rebuttal. I'll keep my previous rating of 5.

---

### Note · Authors · 2025-08-13

We thank all the reviewers for constructive feedback that have ultimately improved the quality of the manuscript. We thank the reviewers for recognizing that our contributions and for pointing that our approach is a "clean, powerful concept" (rFqE), "strongly grounded in a number of theoretical results, and is built from first principles of energy dynamics" (q8MD), "clearly expands the current scope of DAM's" (kb6a). Questions regarding clarification of assumptions, and capacity claims were raised during the review period, which helped improve the manuscript. We are happy to note that a reviewer increased their score in response to our changes. We hope our responses were sufficient to address the concerns of reviewer rFqE. Below we summarize the main changes we make in the paper:

1. Clarification of the Markovian Assumption: The capacity derivation makes use of the Markovian assumption, which we have now detailed in the manuscript.
2. Clarification of the exponential capacity claim: We proved in the manuscript that the capacity is exponential in the number of feature neurons (or state space dimensions). The non-inclusion of hidden neurons in capacity is now made clear in abstract and text. We also note that the definition of capacity follows the currently accepted definitions in the memory literature as evidenced by related works with dense associative memory models.
3. Limitations section. A limitations section has been included that details the assumptions we made in developing our theory and the nuances in the capacity definition.
4. Comparison to related works. Additional comparisons have been included in the introduction section that compares with recent work.

We further thank the reviewers for pointing out the typos in the manuscript, which helped improve its clarity.

---

### Decision · Program_Chairs · 2025-09-17

**Decision:**

Accept (poster)

**Comment:**

Reviewers appreciate the value of the theoretical analysis presented in this paper, which meaningfully extends the results obtained in prior work. In the final camera-ready version authors need to implement the following changes to the manuscript:

1. Make it crystal clear (in the abstract and introduction) that the capacity is exponential in the number of visible neurons, and linear in the number of hidden units.

2. Cite the following two papers and adjust the contribution of their paper accordingly:

     [1] Karuvally, Sejnowski, Siegelmann 2023

     [2] Herron, L., Sartori, P., & Xue, B. (2023). Robust retrieval of dynamic sequences through interaction modulation. PRX Life, 1(2), 023012.

With these modifications in mind, I recommend acceptance.